# Mutual Information Guided Diffusion Model for Partial Label Learning

## Abstract

This paper proposes a novel paradigm for *partial label learning* (PLL) that integrates diffusion mechanisms with *mutual information* (MI) estimation to address the challenge of label disambiguation. In PLL, each training instance is typically associated with multiple candidate labels, among which only one is the ground truth. To simulate the process of label degradation, we introduce noise into the labels through forward diffusion to simulate candidate labels, and then perform reverse denoising to recover the true labels in a probabilistic sense. Unlike in conventional image generation tasks, we present the *Mutual Information Guided Diffusion Model for Partial Label Learning* (MDMPLL), which adapts diffusion models to weakly supervised learning and incorporates MI estimation to strengthen the consistency between denoised labels and data features, thereby improving label disambiguation. Furthermore, *Dual-Path Attention Feature Fusion* (DAFF) strategy is employed to enhance data representation, enabling more effective label disambiguation. Experimental results demonstrate that the proposed method significantly outperforms existing state-of-the-art PLL approaches on multiple datasets.

## 1 Introduction

Modern deep learning models typically rely on large amounts of high-quality annotated data during training. However, obtaining such annotations often requires substantial resources. Moreover, due to subjective differences among annotators, the same image may receive inconsistent labels. Such labeling errors are common in real-world scenarios and can severely affect the learning performance of the model. In this context, the importance of *partial label learning* (PLL) is self-evident. PLL allows each instance to be associated with multiple candidate labels, among which only one is the true class. This not only reduces annotation costs but also enhances model robustness to ambiguous data.

The core of PLL lies in its disambiguation capability for candidate labels. In recent years, researchers have proposed both traditional PLL methods (Luo & Orabona, 2010; Chai et al., 2019) and deep learning-based PLL approaches (Yao et al., 2020a; Fan et al., 2021), have made certain advances. However, existing methods still face several challenges. On the one hand, label ambiguity and the complex relationships among candidate labels make it difficult for some methods to achieve stable disambiguation in a single pass, leading to limited robustness. Many PLL methods rely on single-step predictions for label disambiguation, which is effective in simple cases, but tends to be unstable in environments with significant label ambiguity or noise. Once an error occurs, it is difficult to correct, resulting in limited robustness. On the other hand, label disambiguation and feature learning are often decoupled, with many methods treating these as independent stages, lacking effective mechanisms to associate labels with the semantic content of images. This leads to the inability to synchronize label disambiguation and feature learning during the training process, which limits the performance and stability of the model (Tian et al., 2023; Xia et al., 2023).

To address the above problems, this paper proposes a novel approach called the *Mutual Information Guided Diffusion Model for Partial Label Learning* (MDMPLL). In recent years, diffusion models have demonstrated powerful generative capabilities and achieved remarkable results in fields such as image generation and super-resolution (Starodubcev et al., 2025; Jiang et al., 2025). Inspired by CARD (Han et al., 2022) and DDMP (Fan et al., 2025), this work adapts diffusion models to the

PLL task. We treat the forward noising process as a simulation of label uncertainty, and the reverse denoising process corresponds to label disambiguation. MDMPLL can better model the relationships among candidate labels and the propagation of uncertainty, enabling progressive disambiguation from noisy labels to true ones. This avoids the instability caused by traditional methods that rely on single-step predictions and enhances robustness.

To closely integrate label disambiguation with feature learning and fully leverage multi-level feature information, this paper proposes *Dual-Path Attention Feature Fusion* (DAFF). DAFF combines shallow visual features extracted by an untrained encoder and deep semantic features extracted by a pre-trained encoder, using an attention mechanism to fuse these two types of features. Research shows that shallow and deep features are complementary, with the former focusing on local details and the latter capturing high-level semantic information (Zeiler & Fergus, 2014; Yosinski et al., 2014). This feature fusion effectively improves label disambiguation, with the fused representation serving as a conditional input for label generation, guiding the model to produce label predictions that are more consistent with the semantic content of the images.

Moreover, merely using features as conditional input is insufficient. To encourage the model to retain critical task-relevant information during training and enhance robustness under label disambiguation, MDMPLL introduces a *Feature-Label Mutual Information* (FLMI) maximization mechanism, incorporating it into the training objective as an additional supervisory signal. This mechanism promotes the alignment of features and labels by maximizing the mutual information between the label and its corresponding feature, while minimizing the mutual information between the label and irrelevant features, thereby achieving label disambiguation and improving label prediction accuracy.

MDMPLL also employs a strategy based on *k-nearest neighbors* (KNN) to preprocess ambiguous labels and introduces a dynamic label updating mechanism during training, making the training process more stable. Through iterative forward diffusion and reverse generation, MDMPLL progressively improves its label discrimination ability. The more accurate predicted labels further reinforce the model learning, forming a positive feedback loop.

In summary, the main contributions of this paper are as follows:

- A novel *Mutual Information Guided Diffusion Model for Partial Label Learning* (MDM-PLL) is proposed, redefining partial label learning from a generative modeling perspective to effectively model and disambiguate uncertain labels, thereby achieving improved stability and robustness.

- A *Dual-Path Attention Feature Fusion* (DAFF) module is designed to fully integrate shallow visual features and deep semantic representations. The module leverages attention to highlight discriminative information and serves as conditional input to guide the label disambiguation process.

- A *Feature-Label Mutual Information* (FLMI) maximization mechanism is introduced and incorporated into the training objective, encouraging label representations to retain more task-relevant information from image features, thus enhancing robustness and classification performance under label uncertainty.

## 2 RELATED WORK

The goal of PLL is to identify the ground-truth label from a given candidate label set. Traditional methods typically rely on machine learning approaches such as linear models (Yu & Zhang, 2016), KNN (Zhang & Yu, 2015) and graph-based methods (Lyu et al., 2019), but they often face high computational costs when applied to large-scale data. Deep learning methods have gradually become the mainstream, achieving notable progress via approaches such as self-training (Lv et al., 2020), contrastive learning (Wang et al., 2023) and consistency regularization strategies (Tian et al., 2024). However, existing methods typically over-rely on relationships among candidate labels while neglecting the integration of labels and features, rendering them vulnerable to noisy samples and resulting in unstable and less robust disambiguation.

As a popular class of generative models, diffusion models have been applied to label generation (Han et al., 2022; Chen et al., 2023), yet their application to PLL remains limited, warranting further investigation. Meanwhile, *mutual information* (MI) (Shannon, 1948), a fundamental tool for quantifying

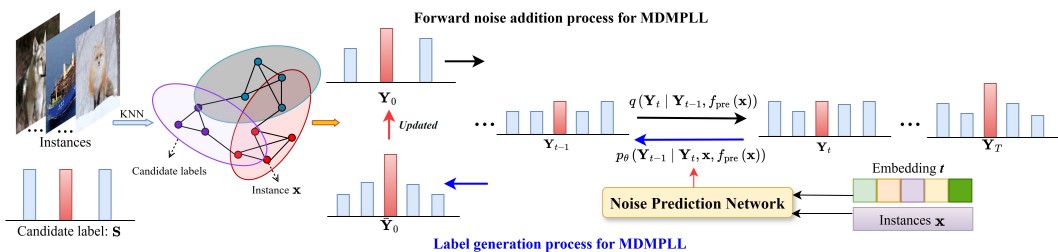

Figure 1: MDMPLL model framework diagram: includes the forward and reverse processes, with noise predicted by the noise prediction network.

the dependency between random variables, has been extensively studied for high-dimensional estimation and has recently been integrated with diffusion models to enhance feature representation and the generation process (Wang et al., 2024; Jiang et al., 2025). This provides a new research direction for PLL, namely combining diffusion models with MI to improve the alignment between labels and image semantics. A more comprehensive review of related work is provided in the Appendix A.

## 3 METHOD

Figure 1 presents the overall framework of the proposed method. First, MDMPLL performs initial disambiguation of candidate labels by leveraging both instance feature similarity and label similarity. Then, the labels undergo a noising process, with the instance and the time embedding $t$ used as conditional inputs to guide label denoising.

### 3.1 PROBLEM FORMULATION

First, we define the PLL problem. Let $\mathcal{X} \in \mathbb{R}^d$ be the feature space and the labeled output space be $\mathcal{Y} = \{1, 2, \ldots, C\}$, where $d$ is the feature dimension and $C$ is the number of classes. The training dataset for PLL is $\mathcal{D} = \{(\mathbf{x}_i, \mathbf{S}_i) \mid 1 \leq i \leq N\}$, where $\mathbf{x}_i \in \mathcal{X}$ denotes the instance features and $\mathbf{S}_i \subseteq \mathcal{Y}$ denotes the candidate labels associated with the instance. We can use $|\mathbf{S}_i|$ to denote the number of candidate labels. The ground-truth label $\mathbf{y}_i$ of the PLL is hidden in its set of candidate labels $\mathbf{S}_i$, and the goal of the model is to learn a classifier $f\colon \mathbf{x}_i \rightarrow \mathbf{y}_i$, that enables it to correctly classify unseen inputs.

### 3.2 FORWARD NOISE ADDITION PROCESS FOR MDMPLL

Inspired by diffusion models (Ho et al., 2020), the model is divided into forward diffusion and backward denoising processes. Before performing forward diffusion on the labels, we disambiguate the initial candidate labels based on the similarity of image features extracted by a pre-trained encoder and the similarity among the labels. Specifically, by computing and combining the structural similarity between samples and the semantic similarity between labels, the original candidate labels are filtered and corrected to obtain more accurate clean labels. This can be expressed by the following equation:

$$\mathbf{Y} = (\mathbf{P} \odot \mathbf{J})\mathbf{S} \tag{1}$$

where $\mathbf{P}$ denotes the feature adjacency matrix, with neighboring features are set to 1. $\mathbf{J}$ is the label similarity matrix, computed using the Jaccard distance. $\odot$ represents the Hadamard product. Since this computation is performed only once at the initial stage of training, the subsequent labels are all denoted by $\mathbf{Y}$.

During the forward diffusion process, the pseudo-clean label $\mathbf{Y}_0$ is gradually perturbed with Gaussian noise to produce a series of noisy labels $\mathbf{Y}_1, \mathbf{Y}_2, \ldots, \mathbf{Y}_T$, eventually forming noise labels that fully follow a Gaussian distribution. The mean of this Gaussian distribution is determined by the features of the instance extracted via the pre-trained encoder $f_{pre}$, which facilitates the generation of labels in the reverse process. The forward process of the diffusion model is essentially a Markov

chain, denoted as $q(\mathbf{Y}_{1:T}|\mathbf{Y}_0, f_{pre}(\mathbf{x}))$. Our diffusion process can be expressed as:

$$q(\mathbf{Y}_{1:T} \mid \mathbf{Y}_0, f_{pre}(\mathbf{x})) = \prod_{t=1}^{T} q(\mathbf{Y}_t \mid \mathbf{Y}_{t-1}, f_{pre}(\mathbf{x})) \tag{2}$$

The single step in which we use the mean and variance can be expressed as:

$$q(\mathbf{Y}_t \mid \mathbf{Y}_{t-1}, f_{pre}(\mathbf{x})) = \mathcal{N}\left(\mathbf{Y}_t; \sqrt{1-\beta_t}\mathbf{Y}_{t-1} + (1-\sqrt{1-\beta_t})f_{pre}(\mathbf{x}), \beta_t\mathbf{I}\right) \tag{3}$$

where $\{\beta_t\}_{t=1:T} \in (0,1)^T$ is a hyperparameter to control the degree of added noise. This formulation allows direct sampling of $\mathbf{Y}_t$, where $t$ can be any time step:

$$q(\mathbf{Y}_t \mid \mathbf{Y}_0, f_{pre}(\mathbf{x})) = \mathcal{N}\left(\mathbf{Y}_t; \sqrt{\bar{\alpha}_t}\mathbf{Y}_0 + (1-\sqrt{\bar{\alpha}_t})f_{pre}(\mathbf{x}), (1-\bar{\alpha}_t)\mathbf{I}\right) \tag{4}$$

where $\alpha_t = 1 - \beta_t$ and $\bar{\alpha}_t = \prod_t \alpha_t$. We reparameterize the sampling of a random variable as a deterministic function plus independent noise, which facilitates gradient computation for the model parameters. After reparameterization, $\mathbf{Y}_t$ can be expressed as:

$$\mathbf{Y}_t = \sqrt{\bar{\alpha}_t}\mathbf{Y}_0 + (1-\sqrt{\bar{\alpha}_t})f_{pre}(\mathbf{x}) + \sqrt{1-\bar{\alpha}_t}\boldsymbol{\epsilon} \tag{5}$$

The full forward diffusion process is presented in the Appendix B.

### 3.3 LABEL GENERATION PROCESS FOR MDMPLL

The reverse process of MDMPLL also follows a Markov chain. We sample from $\mathbf{Y}_T \sim \mathcal{N}(f_{pre}(\mathbf{x}), \mathbf{I})$ to obtain $C$-dimensional Gaussian noise, then perform backward inference via $p_\theta(\mathbf{Y}_{t-1} \mid \mathbf{Y}_t, \mathbf{x}, f_{pre}(\mathbf{x}))$, ultimately recovering the original label vector $\mathbf{Y}_0$. Similar to the forward diffusion, the backward denoising can be expressed as:

$$p_\theta(\mathbf{Y}_{t-1} \mid \mathbf{Y}_t, \mathbf{x}, f_{pre}(\mathbf{x})) = \mathcal{N}\left(\mathbf{Y}_{t-1}; \mu_\theta(\mathbf{Y}_t, \mathbf{x}, f_{pre}(\mathbf{x}), t), \tilde{\beta}_t\mathbf{I}\right) \tag{6}$$

where $\tilde{\beta}_t = \frac{1-\bar{\alpha}_{t-1}}{1-\bar{\alpha}_t} \cdot \beta_t$. The above equation represents the inverse of the noise labeling at step $t-1$ by combining the noise labeling at step $t$ with the conditional input. In the actual calculation, we use the Bayesian formula:

$$q(\mathbf{Y}_{t-1} \mid \mathbf{Y}_t, \mathbf{Y}_0, f_{pre}(\mathbf{x})) \propto q(\mathbf{Y}_t \mid \mathbf{Y}_{t-1}, f_{pre}(\mathbf{x})) \cdot q(\mathbf{Y}_{t-1} \mid \mathbf{Y}_0, f_{pre}(\mathbf{x})) \tag{7}$$

This formula applies the forward noise addition process to the reverse generation. Here, the Bayesian formula is an approximate form of the original, which is easier to compute. By substituting Eq. (3) and Eq. (4) into Eq. (5) and incorporating the idea from DDIM (Song et al., 2020), we can obtain:

$$\begin{aligned}
\mathbf{Y}_{t-1} = &\frac{1}{\sqrt{\alpha_t}}\mathbf{Y}_t + (1 + \bar{\alpha}_{t-1}\sqrt{\alpha_t} - 2\sqrt{\bar{\alpha}_{t-1}}) \cdot f_{pre}(\mathbf{x}) \\
&+ (\sqrt{1-\bar{\alpha}_{t-1}} - \sqrt{\bar{\alpha}_{t-1}}\sqrt{1-\bar{\alpha}_t}) \cdot \boldsymbol{\epsilon}_\theta(\mathbf{Y}_t, \mathbf{x}, f_{pre}(\mathbf{x}), t)
\end{aligned} \tag{8}$$

The only unknown term is $\boldsymbol{\epsilon}_\theta(\mathbf{Y}_t, \mathbf{x}, f_{pre}(\mathbf{x}), t)$, which is what the diffusion model neural network needs to predict and essentially corresponds to the denoising term in the above equation.

In the diffusion model, we aim to maximize the log-likelihood of the training data, $\log p_\theta(\mathbf{Y}_0)$. However, direct maximization is difficult because of the need to integrate over a complex posterior. Therefore, we use variational inference to construct a lower bound, namely the *variational evidence lower bound* (VLB), which serves as our optimization objective:

$$\begin{aligned}
\mathcal{L}_{\text{VLB}} = &\mathbb{E}_q\left[D_{KL}(q(\mathbf{Y}_T|\mathbf{Y}_0, \mathbf{x}, f_{pre})\|p(\mathbf{Y}_T, \mathbf{x}, f_{pre})] + \\
&\sum_{t=2}^{T} \mathbb{E}_q\left[D_{KL}(q(\mathbf{Y}_{t-1}|\mathbf{Y}_t, \mathbf{Y}_0, \mathbf{x}, f_{pre})\|p_\theta(\mathbf{Y}_{t-1}|\mathbf{Y}_t, \mathbf{x}, f_{pre}))] \\
&- \mathbb{E}_q\left[\log p_\theta(\mathbf{Y}_0|\mathbf{Y}_1, \mathbf{x}, f_{pre})]
\end{aligned} \tag{9}$$

The model does not directly optimize this full expression, but replaces it by a simplified surrogate loss:

$$\mathcal{L}_d = \|\boldsymbol{\epsilon} - \boldsymbol{\epsilon}_\theta(\mathbf{Y}_t, \mathbf{x}, f_{pre}(\mathbf{x}), t)\|^2 \tag{10}$$

This MSE loss is a reparameterized approximation of an upper bound term of the VLB, which theoretically guarantees the maximum likelihood property of the model learning process. Predicting the added noise while training the model is more stable and efficient than predicting $\mathbf{Y}_0$ directly, and can be matched to the sampling formula.

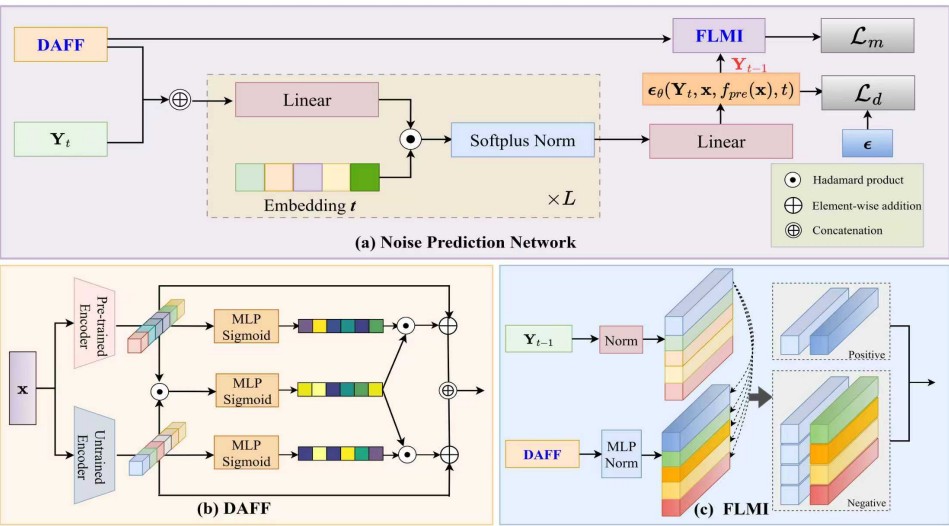

Figure 2: (a) Architecture of the Noise Prediction Network (b) *Dual-Path Attention Feature Fusion* (DAFF) module. (c) *Feature-Label Mutual Information* (FLMI) loss estimation module.

### 3.4 DUAL-PATH ATTENTION FEATURE FUSION

*Multi-source feature fusion* (He et al., 2024; Liu et al., 2024) leverages complementary information across channels or modalities to enhance downstream performance. Traditional concatenation or weighted-sum methods are simple but prone to redundancy and fail to highlight key information. To address this, we propose the *Dual-Path Attention Feature Fusion* (DAFF) module (see Figure 2b), which captures shared information between modalities while preserving branch-specific details.

In DAFF, the instance $\mathbf{x}$ corresponding to the label undergoes two feature extractions: one from an untrained encoder and the other from a pre-trained encoder. Unlike methods that simply multiply the two features, we fuse them through DAFF and then combine the fused representation with the label and time-step embeddings for noise prediction. The untrained features are denoted as $\mathbf{x}_e$, and the pre-trained features as $\mathbf{x}_p$ (directly $f_{pre}(\mathbf{x})$ for large datasets, or fused with the label and time embedding $t$ for small datasets to enhance discriminability). Both $\mathbf{x}_e$ and $\mathbf{x}_p$ are passed through an MLP and a sigmoid activation to obtain their respective attention weights. In addition, we perform element-wise multiplication of the two features and apply the same process to obtain an extra attention weight. This procedure can be formulated as follows:

$$K_{\mathbf{x}_e} = \sigma\left(MLP(\mathbf{x}_e)\right) \quad K_{\mathbf{x}_p} = \sigma\left(MLP(\mathbf{x}_p)\right) \quad K_{\mathbf{x}_c} = \sigma\left(MLP(\mathbf{x}_e \cdot \mathbf{x}_p)\right) \quad (11)$$

where $\sigma$ denotes the sigmoid activation function. $K_{\mathbf{x}_e}$, $K_{\mathbf{x}_p}$ and $K_{\mathbf{x}_c}$ denote the attention weights for $\mathbf{x}_e$, $\mathbf{x}_p$ and their element-wise product, respectively, which are used for weighted fusion. Finally, the attention weights are applied to $\mathbf{x}_e$ and $\mathbf{x}_p$ via element-wise multiplication, and the results are added back to the original features to obtain the fused features $\hat{\mathbf{x}}_e$ and $\hat{\mathbf{x}}_p$. These two features are concatenated along the channel dimension and then passed through a fully connected layer to produce the final feature representation $\mathbf{x}_f$. This process can be formulated as follows:

$$\hat{\mathbf{x}}_e = \mathbf{x}_e + K_{\mathbf{x}_e} \cdot K_{\mathbf{x}_c} \cdot \mathbf{x}_e \quad \hat{\mathbf{x}}_p = \mathbf{x}_p + K_{\mathbf{x}_p} \cdot K_{\mathbf{x}_c} \cdot \mathbf{x}_p \quad \mathbf{x}_f = \text{FC}([\hat{\mathbf{x}}_e; \hat{\mathbf{x}}_p]) \quad (12)$$

where [;] denotes the concatenation operation, and FC($\cdot$) represents a fully connected layer.

DAFF integrates shallow visual and deep semantic features, adaptively emphasizing discriminative information while suppressing irrelevant noise, and capturing their complementary relations to produce more robust and expressive representations for guiding label generation. Theoretically, compared to simple concatenation, DAFF introduces a prior that maximizes FLMI, enabling the fused representation $\mathbf{x}_f$ to achieve a higher mutual information upper bound. The proof can be found in the Appendix D.

---

**Algorithm 1** MDMPLL Framework

---

1: **Input:** Training set $\mathcal{D}$, pre-trained encoder $f_\phi$
2: Obtain the pseudo-clean matrix $\mathbf{Y}$ using Eq. (1)
3: **while** not converged **do**
4:     Sample data $(\mathbf{x}, \mathbf{Y})$
5:     Sample time slice $t \sim \{1, \ldots, T\}$
6:     Sample noise $\epsilon \sim \mathcal{N}(0, \mathbf{I})$
7:     Add noise to $\mathbf{Y}_0$ according to Eq. (5) to obtain $\mathbf{Y}_t$
8:     Sample $\mathbf{Y}_t$ according to Eq. (8) until $t = 0$
9:     Optimize Eq. (16) using gradient descent
10:     Update the candidate labels according to Eq. (17)
11: **end while**

---

### 3.5 Feature-Label Mutual Information

In information theory, the MI between two random variables measures the degree of their dependence. It quantifies how much information about one variable can be obtained by knowing the value of the other. Given two discrete random variables $X$ and $Y$, their MI can be defined as:

$$I(X; Y) = \sum_{\mathbf{x} \in X} \sum_{\mathbf{Y} \in Y} p(\mathbf{x}, \mathbf{Y}) \log \left( \frac{p(\mathbf{x}, \mathbf{Y})}{p(\mathbf{x})p(\mathbf{Y})} \right) \tag{13}$$

where $p(\mathbf{x}, \mathbf{Y})$ is the joint probability of $X$ and $Y$, and $p(\mathbf{x})$ and $p(\mathbf{Y})$ are the marginal probabilities. We estimate the *Feature-Label Mutual Information* (FLMI) using positive and negative samples. As shown in Figure 2c, the label paired with its corresponding feature is treated as a positive sample, while the label paired with other features in the same batch is treated as a negative sample. Here, labels refer to the noise-added labels, and features are the fused representations. The selection of positive and negative samples is defined as follows:

$$L(\mathbf{Y}_t^i, \mathbf{x}_f^j) = \begin{cases} 1, & \text{if } i = j \\ 0, & \text{otherwise} \end{cases} \tag{14}$$

$L$ is the positive and negative sample selection matrix, $\mathbf{Y}_t$ denotes the noisy labels, and $\mathbf{x}_f$ represents the fused features. Noisy labels and fused features help the labels learn from their corresponding features, improving prediction robustness and accuracy. Due to the large dimensional gap between features and labels, we use an MLP to align the feature and label dimensions. Finally, the mutual information loss is derived based on InfoNCE (Kipf et al., 2019). The Appendix E proves that InfoNCE can be used as an estimator of mutual information and enhances label disambiguation performance.

$$\mathcal{L}_m = 1 + \mathbb{E}_P[V(\mathbf{Y}_t, \mathbf{x}_{fp}^+)] - \mathbb{E}_Q[\exp\left(V(\mathbf{Y}_t, \mathbf{x}_{fp}^-)\right)] \tag{15}$$

where $\mathbf{x}_{fp}$ represents the fused features after passing through the MLP. *V* represents the similarity score between two samples.

Figure 2a illustrates the detailed process of the noise prediction network. First, the labels are concatenated with the fused features extracted by DAFF, and the resulting combined representation is first passed through a linear layer and then subjected to a Hadamard product with the time embedding $t$, followed by an activation function and normalization. This sequence of operations is repeated multiple times to produce the final predicted noise representation. The predicted noise is optimized against the ground truth noise using MSE loss. Meanwhile, the fused features and labels are jointly fed into the FLMI module for mutual information estimation, which serves as an additional loss term to further optimize the network. The total loss of our model is:

$$\mathcal{L} = \mathcal{L}_d + \lambda \mathcal{L}_m \tag{16}$$

Finally, we adopt a label update mechanism to prevent the model from drifting in the wrong direction during training. This method can be expressed by the following formula:

$$\mathbf{Y}_0^{k+1} = \left(\mathbf{Y}_0^k + \bar{\mathbf{Y}}_0^k \mathbf{H}\right) \odot \mathbf{m}^k \tag{17}$$

where $k$ denotes the iteration number, $\mathbf{Y}_0^k$ represents the candidate labels after the $k$-th iteration, $\bar{\mathbf{Y}}_0^k$ represents the labels predicted by MDMPLL, and $\mathbf{m}^k$ is the candidate label mask. $\mathbf{H}$ represents

Table 1: Comparative accuracy results (mean ± std) on synthesized benchmark PLL datasets.

| Method | q | MNIST | Fashion-MNIST | Kuzushiji-MNIST | CIFAR-10 | q | CIFAR-100 |
|---|---|---|---|---|---|---|---|
| | | | | Datasets | | | |
| PRODEN | 0.1 | $98.91 \pm 0.03\%$ | $88.27 \pm 0.03\%$ | $91.11 \pm 0.03\%$ | $80.45 \pm 0.24\%$ | 0.01 | $50.42 \pm 0.28\%$ |
| | 0.3 | $98.71 \pm 0.02\%$ | $87.83 \pm 0.04\%$ | $90.67 \pm 0.04\%$ | $79.05 \pm 0.11\%$ | 0.05 | $50.29 \pm 0.29\%$ |
| | 0.5 | $98.57 \pm 0.03\%$ | $87.01 \pm 0.03\%$ | $89.00 \pm 0.05\%$ | $77.52 \pm 0.18\%$ | 0.1 | $46.81 \pm 0.32\%$ |
| LW | 0.1 | $98.76 \pm 0.03\%$ | $88.18 \pm 0.05\%$ | $92.78 \pm 0.04\%$ | $81.43 \pm 0.13\%$ | 0.01 | $51.63 \pm 0.10\%$ |
| | 0.3 | $98.64 \pm 0.02\%$ | $88.02 \pm 0.04\%$ | $91.28 \pm 0.03\%$ | $80.95 \pm 0.17\%$ | 0.05 | $46.54 \pm 0.17\%$ |
| | 0.5 | $98.41 \pm 0.03\%$ | $87.66 \pm 0.06\%$ | $86.55 \pm 0.06\%$ | $78.72 \pm 0.17\%$ | 0.1 | $34.56 \pm 0.45\%$ |
| LS-PLL | 0.1 | $98.83 \pm 0.24\%$ | $90.49 \pm 0.29\%$ | $92.52 \pm 0.37\%$ | $84.31 \pm 0.45\%$ | 0.01 | $53.45 \pm 0.34\%$ |
| | 0.3 | $98.52 \pm 0.36\%$ | $89.78 \pm 0.25\%$ | $86.38 \pm 0.34\%$ | $79.05 \pm 0.52\%$ | 0.05 | $46.23 \pm 0.45\%$ |
| | 0.5 | $98.19 \pm 0.51\%$ | $84.09 \pm 0.56\%$ | $78.85 \pm 0.54\%$ | $67.82 \pm 0.26\%$ | 0.1 | $32.78 \pm 0.74\%$ |
| VALEN | 0.1 | $98.75 \pm 0.02\%$ | $90.72 \pm 0.06\%$ | $92.67 \pm 0.06\%$ | $83.21 \pm 0.10\%$ | 0.01 | $66.79 \pm 0.13\%$ |
| | 0.3 | $98.66 \pm 0.03\%$ | $90.16 \pm 0.06\%$ | $91.75 \pm 0.22\%$ | $82.87 \pm 0.26\%$ | 0.05 | $66.15 \pm 0.25\%$ |
| | 0.5 | $98.32 \pm 0.02\%$ | $89.51 \pm 0.04\%$ | $90.40 \pm 0.03\%$ | $81.88 \pm 0.20\%$ | 0.1 | $65.21 \pm 0.33\%$ |
| PiCO | 0.1 | $98.97 \pm 0.12\%$ | $\mathbf{93.36 \pm 0.09}\%$ | $\mathbf{97.68 \pm 0.06}\%$ | $94.54 \pm 0.05\%$ | 0.01 | $71.02 \pm 0.29\%$ |
| | 0.3 | $98.85 \pm 0.14\%$ | $\mathbf{93.12 \pm 0.12}\%$ | $\mathbf{97.34 \pm 0.03}\%$ | $94.13 \pm 0.08\%$ | 0.05 | $70.29 \pm 0.29\%$ |
| | 0.5 | $98.63 \pm 0.03\%$ | $\mathbf{92.88 \pm 0.03}\%$ | $\mathbf{97.15 \pm 0.12}\%$ | $93.85 \pm 0.15\%$ | 0.1 | $58.16 \pm 0.34\%$ |
| ABLE | 0.1 | $96.72 \pm 0.13\%$ | $92.04 \pm 0.14\%$ | $94.89 \pm 0.03\%$ | $92.98 \pm 0.31\%$ | 0.01 | $52.81 \pm 0.12\%$ |
| | 0.3 | $96.50 \pm 0.52\%$ | $91.91 \pm 0.09\%$ | $94.35 \pm 0.28\%$ | $92.37 \pm 0.15\%$ | 0.05 | $46.26 \pm 0.45\%$ |
| | 0.5 | $96.41 \pm 0.06\%$ | $91.33 \pm 0.17\%$ | $92.98 \pm 0.31\%$ | $91.67 \pm 0.22\%$ | 0.1 | $45.13 \pm 0.21\%$ |
| CAVL | 0.1 | $98.95 \pm 0.04\%$ | $90.32 \pm 0.08\%$ | $93.73 \pm 0.16\%$ | $83.37 \pm 0.07\%$ | 0.01 | $45.80 \pm 0.13\%$ |
| | 0.3 | $98.90 \pm 0.15\%$ | $89.77 \pm 0.04\%$ | $93.57 \pm 0.11\%$ | $81.45 \pm 0.22\%$ | 0.05 | $39.87 \pm 0.33\%$ |
| | 0.5 | $98.71 \pm 0.04\%$ | $88.92 \pm 0.11\%$ | $91.57 \pm 0.22\%$ | $73.25 \pm 0.17\%$ | 0.1 | $21.55 \pm 0.12\%$ |
| CRDPLL | 0.1 | $98.72 \pm 0.22\%$ | $93.21 \pm 0.24\%$ | $97.13 \pm 0.57\%$ | $94.71 \pm 0.18\%$ | 0.01 | $75.31 \pm 0.61\%$ |
| | 0.3 | $98.64 \pm 0.19\%$ | $92.53 \pm 0.31\%$ | $96.55 \pm 0.45\%$ | $94.27 \pm 0.24\%$ | 0.05 | $76.67 \pm 0.18\%$ |
| | 0.5 | $98.32 \pm 0.33\%$ | $91.47 \pm 0.20\%$ | $96.21 \pm 0.61\%$ | $93.84 \pm 0.11\%$ | 0.1 | $71.21 \pm 0.34\%$ |
| LRA-Diffusion | 0.1 | $98.19 \pm 0.11\%$ | $89.96 \pm 0.09\%$ | $95.22 \pm 0.27\%$ | $90.47 \pm 0.13\%$ | 0.01 | $57.53 \pm 0.23\%$ |
| | 0.3 | $97.83 \pm 0.25\%$ | $89.32 \pm 0.24\%$ | $94.73 \pm 0.34\%$ | $89.94 \pm 0.29\%$ | 0.05 | $55.22 \pm 0.23\%$ |
| | 0.5 | $97.59 \pm 0.13\%$ | $86.38 \pm 0.10\%$ | $92.72 \pm 0.16\%$ | $86.86 \pm 0.34\%$ | 0.1 | $49.48 \pm 0.09\%$ |
| MDMPLL (SimCLR) | 0.1 | $98.39 \pm 0.10\%$ | $89.75 \pm 0.12\%$ | $96.01 \pm 0.12\%$ | $90.71 \pm 0.12\%$ | 0.01 | $66.50 \pm 0.18\%$ |
| | 0.3 | $98.27 \pm 0.12\%$ | $89.28 \pm 0.14\%$ | $95.77 \pm 0.06\%$ | $90.47 \pm 0.06\%$ | 0.05 | $65.92 \pm 0.14\%$ |
| | 0.5 | $98.06 \pm 0.03\%$ | $88.49 \pm 0.08\%$ | $94.92 \pm 0.11\%$ | $89.74 \pm 0.05\%$ | 0.1 | $64.65 \pm 0.07\%$ |
| MDMPLL (CLIP) | 0.1 | $\mathbf{99.19 \pm 0.06}\%$ | $92.00 \pm 0.12\%$ | $92.60 \pm 0.05\%$ | $\mathbf{97.71 \pm 0.11}\%$ | 0.01 | $\mathbf{83.95 \pm 0.37}\%$ |
| | 0.3 | $\mathbf{99.00 \pm 0.15}\%$ | $91.43 \pm 0.09\%$ | $91.04 \pm 0.14\%$ | $\mathbf{97.57 \pm 0.15}\%$ | 0.05 | $\mathbf{83.59 \pm 0.17}\%$ |
| | 0.5 | $\mathbf{98.77 \pm 0.11}\%$ | $90.25 \pm 0.12\%$ | $90.44 \pm 0.12\%$ | $\mathbf{97.35 \pm 0.23}\%$ | 0.1 | $\mathbf{83.63 \pm 0.03}\%$ |

the probability distribution over the candidate labels, enabling dynamic updates. The label update process can be regarded as an approximate *expectation-maximization* (EM) algorithm: in the E-step, the diffusion model generates a pseudo-label distribution, in the M-step, the model is updated accordingly. This alternating procedure increases the likelihood under the monotonicity guarantee of EM, further details are provided in the Appendix F. Algorithm 1 outlines the MDMPLL framework.

# 4 EXPERIMENTS

In this section, we demonstrate the effectiveness of MDMPLL through experiments on partial label datasets and compare its performance with existing PLL algorithms.

## 4.1 DATASETS AND BASELINES

To evaluate the effectiveness of our algorithm, we used five commonly adopted benchmark datasets, including CIFAR-10 (Krizhevsky et al., 2009), CIFAR-100 (Krizhevsky et al., 2009), MNIST (Le-Cun et al., 2002), Kuzushiji-MNIST (Clanuwat et al., 2018) and Fashion-MNIST (Xiao et al., 2017). We also included five widely used real-world PLL datasets, namely Lost (Cour et al., 2011), Bird-Song (Briggs et al., 2012), Soccer Player (Liu & Dietterich, 2012), MSRCv2 (Zeng et al., 2013) and Yahoo! News (Guillaumin et al., 2010).

We compared our approach against several state-of-the-art PLL methods, including PRODEN (Yao et al., 2020b), LW (Lv et al., 2020), LS-PLL (Gong et al., 2024), VALEN (Xu et al., 2024), PICO Wang et al. (2023), ABLE (Xia et al., 2022), CAVL (Zhang et al., 2021), CRDPLL (Wu

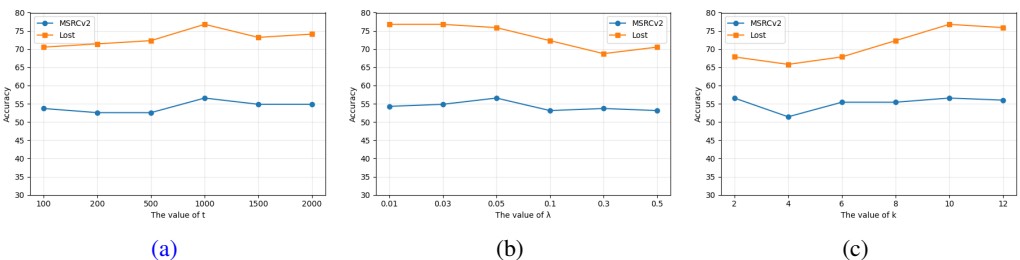

(a)                                    (b)                                    (c)

Figure 3: (a)Accuracy of MDMPLL under mutual information hyperparameter $t$. (b) Accuracy of MDMPLL under mutual information hyperparameter $\lambda$. (c) The accuracy of MDMPLL under $k$ nearest neighbors configuration.

et al., 2022) and LRA-Diffusion (Chen et al., 2023). A detailed description can be found in the Appendix G.

## 4.2 IMPLEMENTATION

We implement MDMPLL in PyTorch and train it for 300 epochs with the Adam optimizer (batch size of 256). Following (Chen et al., 2023), we set the number of sampling trajectories to $s = 10$ and the total number of diffusion steps to $T = 1000$. The number of neighbors in KNN is set to $k = 10$. SimCLR (Chen et al., 2020) and CLIP (Radford et al., 2021) (ViT-L/14) are used as pre-trained encoders, while ResNet serves as the untrained encoder. For real-world partially labeled datasets, we directly employ their preprocessed feature representations.

Table 2: Comparative accuracy results (mean ± std) on real-world PLL datasets.

| Method | Datasets | | | | |
|---|---|---|---|---|---|
| | Lost | MSRCv2 | Birdsong | SoccerPlayer | YahooNews |
| PRODEN | $65.17 \pm 0.72\%$ | $52.00 \pm 0.46\%$ | $63.38 \pm 0.52\%$ | $51.45 \pm 0.19\%$ | $59.06 \pm 0.38\%$ |
| LW | $67.11 \pm 1.78\%$ | $47.15 \pm 0.75\%$ | $66.50 \pm 0.73\%$ | $49.15 \pm 0.38\%$ | $47.03 \pm 0.37\%$ |
| LS-PLL | $66.67 \pm 0.13\%$ | $51.42 \pm 0.38\%$ | $56.80 \pm 0.63\%$ | $54.65 \pm 0.48\%$ | $62.56 \pm 0.35\%$ |
| VALEN | $71.56 \pm 0.80\%$ | $45.89 \pm 0.55\%$ | $72.30 \pm 0.49\%$ | $53.91 \pm 0.12\%$ | $67.73 \pm 0.32\%$ |
| PICO | $65.33 \pm 0.75\%$ | $49.14 \pm 0.57\%$ | $61.29 \pm 2.10\%$ | $55.13 \pm 1.48\%$ | $\mathbf{68.71 \pm 0.22}\%$ |
| ABLE | $68.93 \pm 1.11\%$ | $53.66 \pm 0.95\%$ | $74.70 \pm 0.78\%$ | $\mathbf{62.77 \pm 0.42}\%$ | $53.35 \pm 0.35\%$ |
| CAVL | $63.11 \pm 0.72\%$ | $52.84 \pm 1.16\%$ | $70.50 \pm 0.92\%$ | $54.27 \pm 0.37\%$ | $63.86 \pm 0.58\%$ |
| CRDPLL | $64.55 \pm 0.31\%$ | $49.14 \pm 0.87\%$ | $72.00 \pm 0.62\%$ | $54.47 \pm 0.22\%$ | $65.23 \pm 0.74\%$ |
| LRA-Diffusion | $65.18 \pm 0.24\%$ | $46.86 \pm 0.13\%$ | $63.53 \pm 0.42\%$ | $50.66 \pm 0.27\%$ | $44.45 \pm 0.34\%$ |
| MDMPLL | $\mathbf{76.79 \pm 0.27}\%$ | $\mathbf{56.57 \pm 0.16}\%$ | $\mathbf{75.15 \pm 0.13}\%$ | $60.26 \pm 0.14\%$ | $50.66 \pm 0.26\%$ |

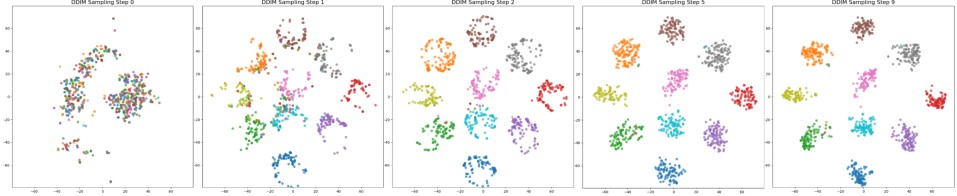

Figure 4: T-SNE visualization of the reverse generation process of MDMPLL on the MNIST dataset.

## 4.3 RESULTS ON SYNTHESIZED AND REAL-WORLD PLL DATASETS

To comprehensively evaluate the effectiveness of MDMPLL, we conducted experiments on multiple synthesized benchmark datasets. As shown in Table 1, MDMPLL significantly outperforms existing methods in most cases. Specifically, MDMPLL (CLIP) achieves improvements of 0.22%, 3.00% and 7.25% over the best baselines on MNIST, CIFAR-10 and CIFAR-100, respectively. Meanwhile, MDMPLL (SimCLR) also achieves competitive or even superior results on most datasets compared

to mainstream approaches. Moreover, MDMPLL maintains strong performance even for large values of $q$, demonstrating strong robustness. These advantages are attributed to the ability of MDMPLL to tightly integrate label disambiguation with feature learning, allowing labels to capture more key feature information and achieving more stable and superior results through a progressive disambiguation strategy. Comparing the two pre-trained models, CLIP generates highly discriminative semantic features with strong feature extraction capability, resulting in more stable and effective performance. However, on unfamiliar datasets, SimCLR may achieve better results.

To further validate the effectiveness of MDMPLL, we evaluated it on multiple real-world partially labeled datasets. As shown in Table 2, MDMPLL demonstrates superior performance on most datasets, achieving higher disambiguation accuracy. On Lost, MSRCv2 and BirdSong, it outperforms the best comparative models by $5.23\%$, $2.91\%$ and $0.45\%$, respectively. On smaller datasets, we did not use pre-trained models, which further highlights the strong performance of MDMPLL. DAFF enriches the effective information in features, FLMI allows labels to fully utilize feature information, and, combined with label preprocessing and update mechanisms, achieves more comprehensive disambiguation.

As shown in Appendix H, MDMPLL reduces label uncertainty by $1/T$ and keeps features and labels tightly coupled, yielding a larger stable-convergence region than discriminative PLL and underpinning the observed robustness. Time complexity analysis is provided in Appendix I.

Table 3: Abalation study of MDMPLL.

| Ablation | D | F | CIFAR-100 | Lost | MSRCv2 | BirdSong |
|---|---|---|---|---|---|---|
| MDMPLL | ✓ | ✓ | $83.59 \pm 0.17\%$ | $76.79 \pm 0.27\%$ | $56.57 \pm 0.16\%$ | $75.15 \pm 0.13\%$ |
| MDMPLL-w/o-D | ✗ | ✓ | $82.25 \pm 0.13\%$ | $71.43 \pm 0.32\%$ | $51.43 \pm 0.23\%$ | $71.74 \pm 0.15\%$ |
| MDMPLL-w/o-F | ✓ | ✗ | $82.84 \pm 0.16\%$ | $74.11 \pm 0.28\%$ | $53.14 \pm 0.09\%$ | $72.95 \pm 0.21\%$ |
| MDMPLL-w/o-DF | ✗ | ✗ | $81.42 \pm 0.21\%$ | $68.75 \pm 0.19\%$ | $50.86 \pm 0.15\%$ | $69.74 \pm 0.26\%$ |

## 4.4 ABLATION STUDY

To evaluate the contributions of each component in the MDMPLL framework, we conducted ablation studies on DAFF and FLMI (Table 3). The results show that removing DAFF (MDMPLL-w/o-D) leads to a performance drop. This indicates that DAFF effectively integrates shallow visual and deep semantic features and extracts key information via the attention mechanism, serving as a conditional input to guide label disambiguation. Removing FLMI (MDMPLL-w/o-F) also reduces performance. This demonstrates that FLMI maximizes the mutual information between features and labels, aligning label embeddings with instance features and preserving more informative features. Simultaneously re-

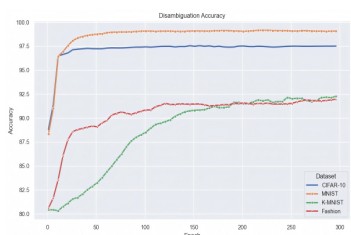

Figure 5: Convergence results of MDMPLL on benchmark datasets.

moving both modules (MDMPLL-w/o-DF) causes a substantial performance decline. This suggests that DAFF and FLMI are closely related. The features obtained through the DAFF module contain sufficient information for the sample's category, enabling FLMI to play a greater role. At the same time, FLMI strengthens the mutual information between labels and features, providing a more stable objective function for the DAFF module, allowing the feature fusion process to better align with the label information.

## 4.5 CONVERGENCE ANALYSIS

To evaluate the performance of MDMPLL, we conducted convergence tests on the CIFAR-10, MNIST, Kuzushiji-MNIST and Fashion-MNIST datasets ($q = 0.3$). As shown in Figure 5, MDMPLL effectively disambiguates labels and converges well on these datasets, demonstrating the stability of its progressive disambiguation and strong overall performance.

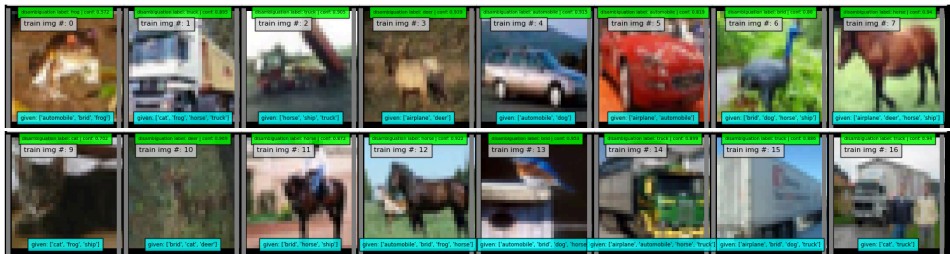

Figure 6: Visualization on the CIFAR-10 dataset with $q = 0.3$.

### 4.6 HYPERPARAMETER ANALYSIS

We analyzed the effect of the time diffusion step $t$ on the MSRCv2 and Lost datasets. As shown in Figure 3a, the best performance is achieved when $t = 1000$, and both higher and lower values of $t$ negatively impact accuracy. We also analyzed the performance of the mutual information hyperparameter $\lambda$ on the MSRCv2 and Lost datasets. As shown in Figure 3b, the model maintains high accuracy even with some variation in $\lambda$, but performance drops significantly when $\lambda$ is too large. Therefore, a recommended range for $\lambda$ is 0.01 to 0.05. We also evaluated the effect of the nearest neighbor parameter $k$. As illustrated in Figure 3c, the model achieves the best performance when $k = 10$. Overall, the analysis indicates that the mutual information loss and label preprocessing significantly enhance the effectiveness of model learning.

### 4.7 VISUALIZATION ANALYSIS

To more intuitively demonstrate the performance of MDMPLL, Figure 4 presents a visualization of the label disambiguation process, where variable step represents the number of reverse sampling steps. As step increases, the labels are gradually denoised, and samples progressively cluster from chaos into distinct groups, restoring the true label distribution and fully illustrating the effectiveness of MDMPLL in label disambiguation. In addition, we visualized the prediction results on the CIFAR-10 dataset ($q = 0.3$) in Figure 6. The blue boxes indicate the candidate label sets for each sample, while the green boxes show the predictions of the model along with their confidence scores. The results demonstrate that MDMPLL can accurately disambiguate labels and maintain high prediction accuracy even in the presence of substantial label noise. More visualizations on other datasets are provided in Appendix J.

## 5 CONCLUSION

This paper proposes MDMPLL, a mutual information guided diffusion model for partial label learning. From the perspective of generative models, it treats the reverse denoising of labels as the core mechanism for disambiguation, enabling stable and progressive label disambiguation. MDMPLL incorporates a DAFF module, which effectively integrates shallow visual features and deep semantic features via an attention mechanism, using them as conditional inputs to guide the disambiguation process. Meanwhile, it introduces a FLMI module to extract shared information between features and labels, thereby tightly coupling feature learning with label disambiguation and further improving disambiguation accuracy. In addition, the framework employs label preprocessing and a dynamic label update mechanism to enhance model stability and generalization. Extensive experiments on multiple partially labeled datasets demonstrate the superior performance of MDMPLL.

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

## A   RELATED WORK

### A.1   PARTIAL LABEL LEARNING

The goal of PLL is to identify the ground-truth label from a given candidate label set. Traditional methods typically rely on machine learning approaches such as linear models (Yu & Zhang, 2016), *k-nearest neighbors* (KNN) (Zhang & Yu, 2015) and graph-based methods (Lyu et al., 2019). However, as data scale increases, these methods often suffer from significantly higher computational complexity. In recent years, deep learning has gradually become the mainstream direction in PLL. Representative works include PRODEN (Lv et al., 2020), which leverages a self-training mechanism to progressively disambiguate labels. RC and CC (Feng et al., 2020) provide theoretical guarantees for risk-consistent and classifier-consistent label selection strategies. PICO (Wang et al., 2023) was the first to introduce contrastive learning loss, laying an important foundation for subsequent research. Crosel (Tian et al., 2024) enhances label identification by adopting a dual-model co-training strategy combined with co-mix consistency regularization. Researchers have also explored structural modeling and attention mechanisms to improve disambiguation accuracy. For instance, KMT-PLL (Fan et al., 2024) integrates K-means clustering with attention mechanisms for more precise label matching. DPLL (Wu et al., 2022) emphasizes learning from non-candidate labels while imposing consistency regularization on candidate labels. Building upon DPLL, CEL (Yang et al., 2025b) leverages class embedding techniques to capture the relationships between candidate and non-candidate labels, further enhancing discriminative ability.

Nevertheless, many PLL methods overly rely on relationships among candidate labels of similar instances, making them vulnerable to noisy neighbors, which may mislead the disambiguation process.

### A.2   DIFFUSION MODEL

In real-world scenarios, researchers not only expect Artificial Intelligence to possess discriminative capabilities but also creative abilities. This demand has led to the emergence of generative models (Kingma & Welling, 2013; Goodfellow et al., 2014; Ho et al., 2020; Trinh & Hamagami, 2024). Among them, diffusion models have gained significant attention in recent years due to their outstanding performance and have been widely applied across various domains (Yuan & Qiao, 2024; Nie et al., 2025). Subsequent studies have introduced labels as guidance in sample generation, thereby achieving effective integration between diffusion models and label information (Rombach et al., 2022; Na et al., 2024). Recently, researchers have extended diffusion model concepts to label generation tasks. For example, CARD (Han et al., 2022) applied diffusion models to classification and regression tasks, marking the first attempt at label generation. LRA (Chen et al., 2023) employed diffusion models to address noisy label problems and achieved promising results.

However, despite their great potential in multiple domains, diffusion models still face major challenges in PLL, where relevant research remains scarce and requires further exploration.

### A.3   MUTUAL INFORMATION

MI (Shannon, 1948) is a fundamental concept in information theory and has been widely applied in unsupervised representation learning to quantify the statistical dependence between random variables. However, estimating MI between high-dimensional variables is challenging. As a result, optimizing and estimating MI have become a critical research direction. Common strategies include maximizing a tractable lower bound or minimizing an upper bound of MI, thereby indirectly optimizing the true MI (Cheng et al., 2020; Hu et al., 2024). These MI estimation methods have been successfully applied across a variety of tasks (Luo et al., 2024; Yang et al., 2025a).

Notably, the integration of MI with diffusion models has recently gained increasing attention. Researchers have explored incorporating MI into the training or generation process of diffusion models to enhance their performance in tasks such as super-resolution reconstruction (Jiang et al., 2025), medical image analysis (Wang et al., 2024) and complex label learning (Li et al.), achieving promising results.

## B DERIVATION OF THE FORWARD PROCESSES IN MDMPLL

The distribution form of the forward diffusion process can be expressed by the following formula:

$$q(\mathbf{Y}_t \mid \mathbf{Y}_{t-1}, f_{pre}(\mathbf{x})) = \mathcal{N}\left(\mathbf{Y}_t; \sqrt{1-\beta_t}\mathbf{Y}_{t-1} + (1-\sqrt{1-\beta_t})f_{pre}(\mathbf{x}), \beta_t\mathbf{I}\right) \quad (18)$$

Here, $f_{pre}(\mathbf{x})$ denotes the mean estimator defined by the encoder.

Let $\alpha_t = 1 - \beta_t$, using the reparameterization trick, we obtain:

$$\mathbf{Y}_t = \sqrt{\alpha_t}\mathbf{Y}_{t-1} + (1-\sqrt{\alpha_t})f_{pre}(\mathbf{x}) + \sqrt{1-\alpha_t}\boldsymbol{\epsilon}_t \quad (19)$$

From this, we can derive the following series of recursive formulas:

$$\mathbf{Y}_1 = \sqrt{\alpha_1}\mathbf{Y}_0 + (1-\sqrt{\alpha_1})f_{pre}(\mathbf{x}) + \sqrt{1-\alpha_1}\boldsymbol{\epsilon}_1$$
$$\mathbf{Y}_2 = \sqrt{\alpha_2}\mathbf{Y}_1 + (1-\sqrt{\alpha_2})f_{pre}(\mathbf{x}) + \sqrt{1-\alpha_2}\boldsymbol{\epsilon}_2$$
$$\cdots \quad (20)$$
$$\mathbf{Y}_{t-1} = \sqrt{\alpha_{t-1}}\mathbf{Y}_{t-2} + (1-\sqrt{\alpha_{t-1}})f_{pre}(\mathbf{x}) + \sqrt{1-\alpha_{t-1}}\boldsymbol{\epsilon}_{t-1}$$
$$\mathbf{Y}_t = \sqrt{\alpha_t}\mathbf{Y}_{t-1} + (1-\sqrt{\alpha_t})f_{pre}(\mathbf{x}) + \sqrt{1-\alpha_t}\boldsymbol{\epsilon}_t$$

By substituting the formula of $\mathbf{Y}_{t-1}$ into that of $\mathbf{Y}_t$, we get:

$$\begin{aligned}\mathbf{Y}_t &= \sqrt{\alpha_t}(\sqrt{\alpha_{t-1}}\mathbf{Y}_{t-2} + (1-\sqrt{\alpha_{t-1}})f_{pre}(\mathbf{x}) \\ &\quad + \sqrt{1-\alpha_{t-1}}\boldsymbol{\epsilon}_{t-1}) + (1-\sqrt{\alpha_t})f_{pre}(\mathbf{x}) + \sqrt{1-\alpha_t}\boldsymbol{\epsilon}_t \\ &= \sqrt{\alpha_t\alpha_{t-1}}\mathbf{Y}_{t-2} + \sqrt{\alpha_t}(1-\sqrt{\alpha_{t-1}})f_{pre}(\mathbf{x}) + (1-\sqrt{\alpha_t}) \\ &\quad f_{pre}(\mathbf{x}) + \sqrt{1-\alpha_t}\boldsymbol{\epsilon}_t + \sqrt{\alpha_t}\sqrt{1-\alpha_{t-1}}\boldsymbol{\epsilon}_{t-1}\end{aligned} \quad (21)$$

Since both $\boldsymbol{\epsilon}_{t-1}$ and $\boldsymbol{\epsilon}_t$ follow the Gaussian distribution $\mathcal{N}(0,1)$, they can be simplified using the addition formula for Gaussian functions. After simplification, we obtain:

$$\mathbf{Y}_t = \sqrt{\alpha_t\alpha_{t-1}}\mathbf{Y}_{t-2} + (1-\sqrt{\alpha_t\alpha_{t-1}})f_{pre}(\mathbf{x}) + \sqrt{1-\alpha_t\alpha_{t-1}}\boldsymbol{\epsilon} \quad (22)$$

By recursively substituting the equations, we can ultimately derive the relationship between $\mathbf{Y}_t$ and $\mathbf{Y}_0$:

$$\mathbf{Y}_t = \sqrt{\alpha_t\alpha_{t-1}\cdots\alpha_1}\mathbf{Y}_0 + (1-\sqrt{\alpha_t\alpha_{t-1}\cdots\alpha_1})f_{pre}(\mathbf{x})$$
$$+\sqrt{1-\alpha_t\alpha_{t-1}\cdots\alpha_1}\boldsymbol{\epsilon} \quad (23)$$

By setting $\bar{\alpha}_t = \alpha_t\alpha_{t-1}\cdots\alpha_1$, we finally obtain the well-known forward diffusion formula:

$$\mathbf{Y}_t = \sqrt{\bar{\alpha}_t}\mathbf{Y}_0 + (1-\sqrt{\bar{\alpha}_t})f_{pre}(\mathbf{x}) + \sqrt{1-\bar{\alpha}_t}\boldsymbol{\epsilon} \quad (24)$$

## C DERIVATION OF PLL FROM A GENERATIVE MODELING PERSPECTIVE

In PLL, the training data consists of input features $\mathbf{x} \in \mathbb{R}^d$ and a candidate label set $\mathbf{S} \subseteq \mathcal{Y} = \{1, 2, \ldots, C\}$. Each candidate label set contains an unobserved true label $\mathbf{y} \in \mathbf{S}$. Before training begins, we perform a one-time preprocessing step to freeze $\mathbf{S}$ into the initial candidate-label matrix $\mathbf{Y}$, and subsequently write $\mathbf{y} \in \mathbf{Y}$ for notational consistency. Our goal is to learn the conditional distribution $p(\mathbf{y} \mid \mathbf{x})$ to predict the true label.

From the generative perspective, the candidate label set $\mathbf{Y}$ is considered as an intermediate variable generated from the true label $\mathbf{y}$, whose generation process depends on the latent true label and the input features. We aim to model the joint probability of the sample $\mathbf{x}$ and the candidate label set $\mathbf{Y}$ to derive the formulation. We start by modeling the marginal likelihood $p(\mathbf{Y} \mid \mathbf{x})$. Since the true label $\mathbf{y}$ is an unobserved latent variable, we marginalize it out:

$$p(\mathbf{Y} \mid \mathbf{x}) = \sum_{\mathbf{y}\in\mathcal{Y}} p(\mathbf{Y}, \mathbf{y} \mid \mathbf{x}) \quad (25)$$

By the chain rule of conditional probability, we have:

$$p(\mathbf{Y} \mid \mathbf{x}) = \sum_{\mathbf{y} \in \mathcal{Y}} p(\mathbf{Y} \mid \mathbf{y}, \mathbf{x}) \cdot p(\mathbf{y} \mid \mathbf{x}) \tag{26}$$

To simplify modeling, we make the following assumption:

**Assumption**: The generation of the candidate label set depends only on the true label and is independent of the input features $\mathbf{x}$, i.e.,

$$p(\mathbf{Y} \mid \mathbf{y}, \mathbf{x}) = p(\mathbf{Y} \mid \mathbf{y}) \tag{27}$$

Substituting this into Eq. (25), we obtain:

$$p(\mathbf{Y} \mid \mathbf{x}) = \sum_{\mathbf{y} \in \mathcal{Y}} p(\mathbf{Y} \mid \mathbf{y}) \cdot p(\mathbf{y} \mid \mathbf{x}) \tag{28}$$

By definition of PLL, the candidate label set $\mathbf{Y}$ must contain the true label $\mathbf{y}$, i.e., $\mathbf{y} \in \mathbf{Y}$. Therefore, we restrict the summation to $\mathbf{y} \in \mathbf{Y}$:

$$p(\mathbf{Y} \mid \mathbf{x}) = \sum_{\mathbf{y} \in \mathbf{Y}} p(\mathbf{Y} \mid \mathbf{y}) \cdot p(\mathbf{y} \mid \mathbf{x}) \tag{29}$$

## D  THEORETICAL JUSTIFICATION FOR DAFF IMPROVING MUTUAL INFORMATION ESTIMATION

Let $\mathbf{x}_e$ be shallow visual features, $\mathbf{x}_p$ be deep semantic features, $\mathbf{x}_f = \text{DAFF}(\mathbf{x}_e, \mathbf{x}_p)$ be the fused representation. If the fusion function $\text{DAFF}(\cdot)$ is a *sufficient statistic* of $(\mathbf{x}_e, \mathbf{x}_p)$ for the label $\mathbf{Y}$, i.e. the Markov chain $\mathbf{Y} \rightarrow (\mathbf{x}_e, \mathbf{x}_p) \rightarrow \mathbf{x}_f$ holds, then the InfoNCE estimator satisfies:

$$I(\mathbf{x}_f; \mathbf{Y}) \geq \max\{I(\mathbf{x}_e; \mathbf{Y}), I(\mathbf{x}_p; \mathbf{Y})\} \tag{30}$$

**Proof.** InfoNCE loss lower-bounds mutual information, for any feature $\mathbf{z}$, we have:

$$I(\mathbf{z}; \mathbf{Y}) = \mathbb{E}\left[\log \frac{e^{s(\mathbf{z}, \mathbf{Y})}}{\frac{1}{K}\sum_{j=1}^{K} e^{s(\mathbf{z}, \mathbf{Y}_j)}}\right] + \log K \tag{31}$$

where $s(\mathbf{z}, \mathbf{Y}) = \phi(\mathbf{z})^T \psi(\mathbf{Y})$ is a similarity score, $\phi$ and $\psi$ are learnable networks, and $\{\mathbf{Y}_j\}_{j=1}^{K}$ contains one positive and $K - 1$ negatives sampled from the marginal distribution.

Because $\mathbf{x}_f$ is a deterministic function of $(\mathbf{x}_e, \mathbf{x}_p)$, the data-processing inequality gives

$$I(\mathbf{x}_f; \mathbf{Y}) \leq I(\mathbf{x}_e, \mathbf{x}_p; \mathbf{Y}) \tag{32}$$

However, InfoNCE is *a lower bound* whose tightness depends on how well the representation separates the positive pair from the negatives. DAFF uses an attention mechanism to *jointly exploit* complementary cues in $\mathbf{x}_e$ and $\mathbf{x}_p$, yielding larger expected similarity for the true pair and smaller similarities for negatives. Formally,

$$\mathbb{E}\left[\log \sum_{j=1}^{K} e^{s(\mathbf{x}_f, \mathbf{Y}_j)}\right] \leq \min\left\{\mathbb{E}\left[\log \sum_{j=1}^{K} e^{s(\mathbf{x}_e, \mathbf{Y}_j)}\right], \mathbb{E}\left[\log \sum_{j=1}^{K} e^{s(\mathbf{x}_p, \mathbf{Y}_j)}\right]\right\} \tag{33}$$

Substituting this inequality into the InfoNCE expression directly implies

$$I(\mathbf{x}_f; \mathbf{Y}) \geq \max\{I(\mathbf{x}_e; \mathbf{Y}), I(\mathbf{x}_p; \mathbf{Y})\} \tag{34}$$

DAFF produces a fused representation that attains a *tighter InfoNCE lower bound* than either shallow or deep features alone, thereby increasing the estimated mutual information between features and labels and improving label-disambiguation performance.

# E   INFONCE LOSS AND MI ESTIMATION

## E.1   INTRODUCTION TO INFONCE LOSS

*Information noise contrastive estimation* (InfoNCE) is a loss function commonly used in contrastive learning. It aims to maximize the similarity between positive sample pairs while minimizing the similarity between positive and negative sample pairs. This loss is primarily used in unsupervised representation learning to estimate a lower bound of MI.

Given a positive sample pair $(\mathbf{x}, \mathbf{Y}^+)$ and $K - 1$ negative sample pairs $(\mathbf{x}, \mathbf{Y}_i^-)$, the InfoNCE loss is defined as:

$$\mathcal{L}_{\text{InfoNCE}} = -\mathbb{E}\left[\log \frac{f(\mathbf{x}, \mathbf{Y}^+)}{f(\mathbf{x}, \mathbf{Y}^+) + \sum_{i=1}^{K-1} f(\mathbf{x}, \mathbf{Y}_i^-)}\right] \tag{35}$$

where $f(\mathbf{x}, \mathbf{Y}')$ is a scoring function that measures similarity, typically an exponentiated similarity such as $\exp(\text{sim}(\mathbf{x}, \mathbf{Y}')/\tau)$, with $\tau$ being a temperature parameter.

## E.2   INFONCE IS A LOWER BOUND OF MI

According to (Oord et al., 2018), InfoNCE actually provides a trainable lower bound of the MI $I(X; Y)$. Let $(\mathbf{x}, \mathbf{Y})$ be a positive sample pair, and $\{\mathbf{Y}_i\}_{i=1}^{K-1}$ be negative samples drawn from the marginal distribution $p(\mathbf{Y})$. Define $f(\mathbf{x}, \mathbf{Y}) = \frac{p(\mathbf{Y}|\mathbf{x})}{p(\mathbf{Y})}$, the ratio between the posterior and marginal distributions. It can be proved that:

$$I(X; Y) \geq \log K - \mathcal{L}_{\text{InfoNCE}} \tag{36}$$

where $K$ is the total number of samples consisting of one positive sample and $K - 1$ negative samples.

The detailed derivation is as follows: Let the joint distribution be $p(\mathbf{x}, \mathbf{Y})$, and the marginal distribution be $p(\mathbf{x})p(\mathbf{Y})$. The MI is defined as:

$$\begin{aligned} I(X; Y) &= \mathbb{E}_{p(\mathbf{x}, \mathbf{Y})}\left[\log \frac{p(\mathbf{x}, \mathbf{Y})}{p(\mathbf{x})p(\mathbf{Y})}\right] \\ &= \mathbb{E}_{p(\mathbf{x})}\left[D_{\text{KL}}\big(p(\mathbf{Y} \mid \mathbf{x}) \parallel p(\mathbf{Y})\big)\right] \end{aligned} \tag{37}$$

InfoNCE essentially approximates the KL divergence by a $K$-way softmax classification task:

$$\mathcal{L}_{\text{InfoNCE}} = \mathbb{E}\left[-\log \frac{e^{s(\mathbf{x}, \mathbf{Y})}}{e^{s(\mathbf{x}, \mathbf{Y})} + \sum_{i=1}^{K-1} e^{s(\mathbf{x}, \mathbf{Y}_i)}}\right] \tag{38}$$

When the scoring function is set as $s(\mathbf{x}, \mathbf{Y}) = \log \frac{p(\mathbf{Y}|\mathbf{x})}{p(\mathbf{Y})}$, we have:

$$\begin{aligned} \mathcal{L}_{\text{InfoNCE}} &= -\mathbb{E}_{p(\mathbf{x}, \mathbf{Y})}\left[\log \frac{\frac{p(\mathbf{Y}|\mathbf{x})}{p(\mathbf{Y})}}{\frac{p(\mathbf{Y}|\mathbf{x})}{p(\mathbf{Y})} + \sum_{i=1}^{K-1} \frac{p(\mathbf{Y}_i)}{p(\mathbf{Y}_i)}}\right] \\ &= -\mathbb{E}_{p(\mathbf{x}, \mathbf{Y})}\left[\log \frac{\frac{p(\mathbf{Y}|\mathbf{x})}{p(\mathbf{Y})}}{\frac{p(\mathbf{Y}|\mathbf{x})}{p(\mathbf{Y})} + K - 1}\right] \end{aligned} \tag{39}$$

From the above, we obtain the following lower bound:

$$I(X; Y) \geq \log K - \mathcal{L}_{\text{InfoNCE}} \tag{40}$$

## E.3   PRACTICAL APPLICATION

In practical contrastive learning or semi-supervised learning tasks, we usually do not explicitly construct the distributions $p(\mathbf{Y} \mid \mathbf{x})$ and $p(\mathbf{Y})$. Instead, a scoring function $f(\mathbf{x}, \mathbf{Y})$ is learned via neural networks. Although $f(\mathbf{x}, \mathbf{Y})$ is no longer exactly proportional to $\frac{p(\mathbf{Y}|\mathbf{x})}{p(\mathbf{Y})}$, InfoNCE can still be regarded as maximizing a lower bound of the MI $I(X; Y)$. Therefore, it can be used for unsupervised feature learning and MI estimation.

### E.4 InfoNCE Maximization Improves Label Disambiguation

In PLL each sample $\mathbf{x}$ is associated with a pre-processed candidate label vector $\mathbf{Y}$. Maximizing $I(X;Y)$ minimizes the conditional entropy $H(\mathbf{Y}|\mathbf{x})$ over $\mathbf{Y}$, sharpening the posterior $p_\theta(\mathbf{y}|\mathbf{x})$.

For any $\mathbf{x}$, $\arg\min H(\mathbf{Y}|\mathbf{x})$ concentrates all mass on the single true label inside $\mathbf{Y}$.

$H(\mathbf{Y}|\mathbf{x}) = -\sum_{\mathbf{y}\in\mathbf{Y}} p_\theta(\mathbf{y}|\mathbf{x}) \log p_\theta(\mathbf{y}|\mathbf{x})$. The uniform distribution maximizes entropy, the one-hot distribution minimizes it. Hence $\arg\min H(\mathbf{Y}|\mathbf{x})$ is the one-hot vector on the ground-truth label.

Since our FLMI loss is exactly $-\text{InfoNCE}$ and $\text{InfoNCE} \geq \log|\mathbf{Y}| - H(\mathbf{Y}|\mathbf{x})$, minimizing FLMI drives $H(\mathbf{Y}|\mathbf{x}) \to 0$, forcing $p_\theta(\mathbf{y}|\mathbf{x})$ to collapse onto the unique correct label within $\mathbf{Y}$, thereby performing label disambiguation.

## F Derivation of Label Update from an EM Perspective

The update in Eq (17) can be interpreted as an EM-style algorithm to maximize the likelihood of observed data.

$$\mathcal{G}(\theta) = \prod_{\mathbf{x}} \sum_{\mathbf{y}\in\mathbf{Y}} p_\theta(\mathbf{y}|\mathbf{x}) \tag{41}$$

**E-step:** compute the *responsibility*

$$q^{k+1}(\mathbf{y}|\mathbf{x}) = \frac{\mathbf{H}(\mathbf{x},\mathbf{y})\, p_{\theta^k}(\mathbf{y}|\mathbf{x})}{Z(\mathbf{x})} \qquad Z(\mathbf{x}) = \sum_{\mathbf{y}'\in\mathbf{Y}} \mathbf{H}(\mathbf{x},\mathbf{y}')\, p_{\theta^k}(\mathbf{y}'|\mathbf{x}) \tag{42}$$

where $\mathbf{H}(\mathbf{x},\mathbf{y})$ represents the probability distribution over the candidate labels, hence $q^{k+1}$ is a valid distribution over the candidate set $\mathbf{Y}$. Here, $\mathbf{y}'$ is a dummy variable introduced for summation over the candidate set $\mathbf{Y}$, it ranges over the same values as $\mathbf{y}$ and is used only in the computation of the normalization constant $Z(\mathbf{x})$.

**M-step:** maximize the *expected complete-data log-likelihood*

$$\mathcal{Q}(\theta|\theta^k) = \sum_{\mathbf{x}} \sum_{\mathbf{y}\in\mathbf{Y}} q^{k+1}(\mathbf{y}|\mathbf{x}) \log p_\theta(\mathbf{y}|\mathbf{x}) \tag{43}$$

Since the diffusion training objective is exactly the negative log-likelihood of $p_\theta(\mathbf{y}|\mathbf{x})$, one stochastic-gradient step using labels sampled from $q^{k+1}$ already ensures a non-decreasing $\mathcal{Q}(\theta|\theta^k)$, exact maximization is not required for monotonicity, thus our method is a *Generalized EM* (GEM) rather than exact EM.

**Monotonicity:** Standard EM (and GEM) theory gives

$$\mathcal{G}(\theta^{k+1}) \geq \mathcal{G}(\theta^k) \tag{44}$$

with equality iff $\theta^k$ is a stationary point of $\mathcal{G}(\theta)$. Therefore the sequence $\{\theta^k\}$ converges to a stationary point of the observed-data likelihood.

Table 4: Detailed information of real-world PLL datasets, including the number of features (Fea), number of instances (Ins), mean number of ambiguous labels in the candidate label set (MEA), total number of labels (Labels) and application scenario (Scenario).

| Dataset | Fea | Ins | MEA | Labels | Scenario |
|---|---|---|---|---|---|
| MSRCv2 | 48 | 1,758 | 3.16 | 23 | Object Classification |
| BirdSong | 38 | 4,998 | 2.18 | 13 | Bird Song Classification |
| Lost | 108 | 1,122 | 2.33 | 16 | Automatic Face Naming |
| Yahoo! News | 163 | 22,991 | 1.91 | 219 | Automatic Face Naming |
| SoccerPlayer | 279 | 17,472 | 2.09 | 171 | Automatic Face Naming |

## G DETAILED DESCRIPTION OF DATASETS AND BASELINES

### G.1 DATASETS

To evaluate the effectiveness of the proposed algorithm, we employ five commonly used benchmark datasets: CIFAR-10, CIFAR-100, MNIST, Kuzushiji-MNIST and Fashion-MNIST. These datasets are converted into Synthesized PLL datasets using the standard synthesis procedure. Specifically, for each instance, the candidate label set is generated by independently selecting Q-1 noisy labels, where the parameter q controls the probability of each label being chosen as a noise label. Generally, a higher q value leads to more noise labels being included in the candidate set, thereby increasing the difficulty of label disambiguation. For MNIST, Kuzushiji-MNIST, Fashion-MNIST and CIFAR-10, we consider $q \in \{0.1, 0.3, 0.5\}$, for CIFAR-100, we use $q \in \{0.01, 0.05, 0.1\}$.

To comprehensively and effectively evaluate the performance of the proposed MDMPLL method, we also adopt five widely used real-world PLL datasets: Lost, BirdSong, Soccer Player, MSRCv2 and Yahoo! News. These datasets span a variety of application scenarios, such as automatic face naming, object classification and bird song classification. The detailed information of these real-world PLL datasets is summarized in Table 4.

### G.2 COMPARED METHODS

To verify the superiority of the proposed MDMPLL method, we conducted comparative experiments with several state-of-the-art PLL methods. Below is a brief introduction of each method:

- **PRODEN** (Yao et al., 2020b): Employs collaborative training of multiple networks and a progressive label disambiguation strategy to dynamically update label probabilities, boosting model discrimination capability.

- **LW** (Lv et al., 2020): Combines sample confidence estimation with optimal transport theory to adjust label distribution weights, enhancing robustness against noisy labels and disambiguation performance.

- **LS-PLL** (Gong et al., 2024): Proposes a label-smoothed deep partial-label learning framework, derives the optimal smoothing rate theoretically, and designs a practical algorithm that significantly boosts robustness.

- **VALEN** (Xu et al., 2024): Learns personalized label distributions for each instance via variational inference, enhancing modeling of instance-dependent labels and improving disambiguation accuracy.

- **PiCO** (Wang et al., 2023): Introduces a contrastive-learning mechanism that refines prototype representations to strengthen feature discrimination and achieve more robust label disambiguation.

- **ABLE** (Xia et al., 2022): Applies a contrastive learning strategy driven by label ambiguity, explicitly leveraging ambiguous label information to enhance feature representation.

- **CAVL** (Zhang et al., 2021): Utilizes Class Activation Maps to guide the model attention toward discriminative regions, boosting the accuracy of true-label recognition through activation values.

- **CRDPLL** (Wu et al., 2022): Employs consistency regularization to improve output stability under input perturbations, enhancing generalization and label disambiguation.

- **LRA-Diffusion** (Chen et al., 2023): Introduces a label-retrieval-augmented diffusion model that improves learning from noisy labels by leveraging label retrieval techniques, enhancing the model's robustness to noisy label scenarios.

Parameter settings for each method follow those specified in their original papers.

# H  THEORETICAL GUARANTEES ON ROBUSTNESS AND FEATURE–LABEL COUPLING

MDMPLL integrates the Dual-Path Attention Feature Fusion (DAFF) module with multi-step reverse diffusion and mutual information maximization to achieve robust learning under partial label noise.

**DAFF Feature Fusion.** For each instance, DAFF fuses shallow visual features and deep semantic features to generate a discriminative representation $\mathbf{x}_f$. By adaptively emphasizing informative components and suppressing irrelevant noise, DAFF produces robust feature embeddings that capture complementary information from both feature levels. Maximizing the InfoNCE lower bound,

$$I(\mathbf{x}_f; \mathbf{Y}) \geq \log K - \mathcal{L}_{\text{InfoNCE}}, \tag{45}$$

ensures that the fused representation remains tightly coupled with labels, suppressing the contribution of mislabeled or ambiguous candidates.

**Multi-Step Diffusion.** Each reverse diffusion step updates pseudo-labels via a weighted average of neighbors in feature space. Assuming the averaging weights sum to one and the updates are approximately independent with bounded variance, the expected label uncertainty decreases at a rate of $1/T$. Multi-step updates progressively refine pseudo-labels, suppress fluctuations, and stabilize the learned feature–label associations, reinforcing the robustness provided by DAFF.

**Overall Objective.** The training objective of MDMPLL,

$$\min_\theta \mathbb{E}[\|\boldsymbol{\epsilon} - \boldsymbol{\epsilon}_\theta(\mathbf{Y}_t, \mathbf{x}, f_{pre}(\mathbf{x}), t)\|^2] - \lambda I(\mathbf{x}_f; \mathbf{Y}) \tag{46}$$

combines diffusion reconstruction with feature–label mutual information maximization. DAFF ensures that $\mathbf{x}_f$ is highly informative, multi-step diffusion reduces label uncertainty, and InfoNCE maintains tight feature–label coupling.

Collectively, these mechanisms reduce label uncertainty through iterative pseudo-label refinement, stabilize feature–label associations by leveraging the discriminative fused representation from DAFF, and prevent the propagation of errors from noisy or ambiguous labels. This integrated design expands the stable-convergence region of MDMPLL, improves the reliability of downstream predictions, and underpins the observed robustness compared to single-step discriminative PLL.

This integrated design expands the stable-convergence region of MDMPLL, improves the reliability of downstream predictions, and underpins the observed robustness compared to single-step discriminative PLL.

# I  TIME COMPLEXITY ANALYSIS

We analyse one gradient step of MDMPLL. Let $N$ denote the batch size, $C$ the number of classes, $d$ the feature dimension, $T$ the number of reverse diffusion steps, and $W$ the approximate per-forward computational cost of the noise-prediction network (used as a proxy for FLOPs or parameter-dependent cost).

The computational components per step are as follows. Feature extraction via the frozen encoder costs $\mathcal{O}(Nd)$, which can be amortized if features are cached. The noise-prediction network is executed $T$ times, giving a cost of $\mathcal{O}(TNW)$ per-sample (or $\mathcal{O}(TCW)$ if evaluated per class per step). InfoNCE similarity computation forms the $N \times C$ matrix, costing $\mathcal{O}(NCd)$. DAFF feature fusion involves two lightweight MLPs and element-wise operations on $N \times d$ tensors, contributing $\mathcal{O}(Nd)$, which is dominated by the InfoNCE term.

Hence, under the common per-sample evaluation assumption, one training step costs $\mathcal{O}(TNW + NCd)$, and after feature caching (removing the $\mathcal{O}(Nd)$ encoder cost) the dominant terms are the two above. In our implementation we use DDIM with $s = T = 10$.

We evaluated MDMPLL as well as several comparison methods on an NVIDIA RTX 4090 GPU. On the CIFAR-10 dataset (with $q = 0.3$), the training time for MDMPLL is approximately 4.0 GPU hours, while PICO takes about 3.2 GPU hours, CAVL requires about 2.6 GPU hours, and LS-PLL takes about 2.8 GPU hours. Despite the introduction of iterative diffusion steps, MDMPLL

achieves stable convergence within 300 epochs. Moreover, unlike traditional image diffusion models, MDMPLL adds and denoises noise on the labels rather than on the pixels. This design allows our noise prediction network to remain lightweight and efficient, thereby accelerating the training speed for each epoch. Therefore, although MDMPLL incurs higher computational costs, it achieves significantly higher accuracy, making the additional complexity both reasonable and worthwhile.

## J VISUALIZATION ANALYSIS OF THE PREDICTION RESULTS

This subsection presents the visualization of prediction results on CIFAR-100, Fashion-MNIST, MNIST and Kuzushiji-MNIST. Among them, Fashion-MNIST, MNIST and Kuzushiji-MNIST exhibit relatively high confidence. For CIFAR-100, although the confidence is lower due to the large number of classes, it still maintains a high prediction accuracy.

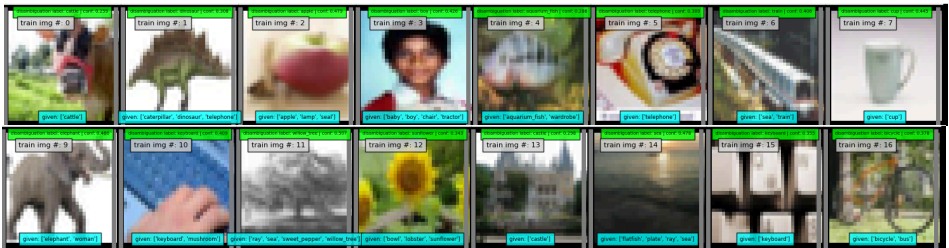

Figure 7: Visualization on the CIFAR-100 dataset with $q = 0.05$.

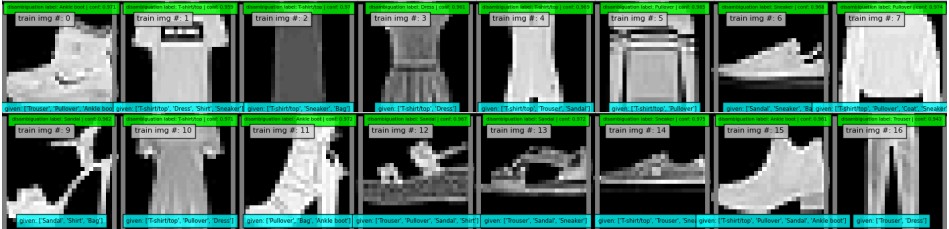

Figure 8: Visualization on the Fashion-MNIST dataset with $q = 0.3$.

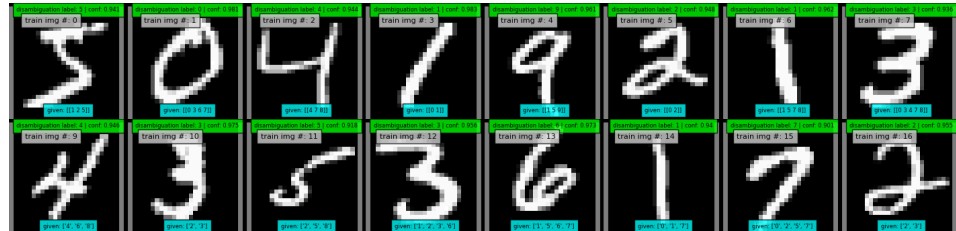

Figure 9: Visualization on the MINST dataset with $q = 0.3$.

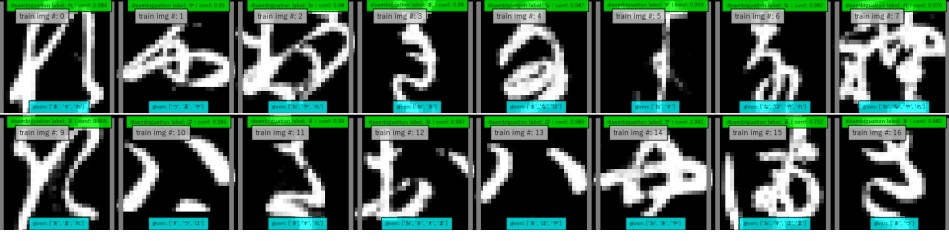

Figure 10: Visualization on the Kuzushiji-MNIST dataset with $q = 0.3$.

## K    THE USE OF LLMS

During the writing of this paper, we used a large language model (e.g., ChatGPT) to aid writing and text polishing. All research ideas, experiments, and results were independently completed by the authors.

