# OpenReview forum: "Mutual Information Guided Diffusion Model for Partial Label Learning"
_ICLR.cc/2026/Conference — Submitted to ICLR 2026_

### Official Review · Reviewer_Zzau · 2025-10-27

**Soundness:** 2
**Presentation:** 3
**Contribution:** 3
**Rating:** 4
**Confidence:** 4

**Summary:**

This paper investigates the problem of partial-label learning. It leverages the reverse denoising process of diffusion models to disambiguate partial-label data. To encourage the model to retain task-relevant information during training and to enhance robustness under ambiguous labels, the paper introduces a feature–label mutual information maximization mechanism and incorporates it into the training objective as an additional supervisory signal.

**Strengths:**

1.	The paper has a strong motivation, applying the denoising process of diffusion models to label disambiguation in partial-label learning.
2.	The writing is well organized and adheres to academic conventions.

**Weaknesses:**

1.	The paper’s review of existing methods is overly abstract and insufficiently specific. For example: “On the one hand, label ambiguity and the complex relationships among candidate labels make it difficult for some methods to achieve stable disambiguation in a single pass, leading to limited robustness.” Nearly all PLL methods aim to address precisely this issue, so this statement does not accurately pinpoint the concrete shortcomings of prior work.
2.	Regarding “On the other hand, label disambiguation and feature learning are often decoupled, and lack mechanisms that effectively associate labels with the semantic content of images.” The intended claim seems to be that disambiguation and representation learning are decoupled and lack a mechanism to link them for stable, mutually reinforcing training. However, the proposed DAFF targets only representation learning and does not provide a joint mechanism. Moreover, many existing PLL approaches already consider disambiguation and representation learning jointly rather than as two separate stages.
3.	The paper does not explain why diffusion models are suitable for label disambiguation in PLL, which is crucial. Since the label information fed into the diffusion model is noisy, if the denoising process outputs a noisy label at each step, can it ultimately recover the true label? This justification is missing.

**Questions:**

1.	Are there any fundamental issues with applying diffusion models to partial-label learning? Why is $Y_0$ initialized via disambiguation from similar examples (KNN), and how can label information that is still noisy be used as input to a diffusion model? In image denoising, the diffusion model’s initial input is a clean, fully observed image. When applying diffusion to PLL, the input labels are noisy. Why does using KNN-based initialization improve the stability of disambiguation? What problems arise if it is not used?
2.	The proposed DAFF module appears to act only on representation learning and does not resolve the decoupling between disambiguation and representation learning found in prior methods.
3.	From the ablation results, the contributions of components D and F are modest, yet the overall gain over existing methods is large. This suggests the diffusion model itself contributes substantially. It would be better to show ablations across multiple datasets.
4.	After training, can the model use the diffusion denoising process to directly predict labels for unseen examples?
5.	In Eq. (15), what does $𝑉$ denote? What is the relationship between Eq. (9) and Eqs. (10), (15), and (16)?

---

> ### Author Response · Authors · 2025-11-20
>
> Dear Reviewer Zzau,
>
> Thank you for your meticulous review and valuable feedback on our paper. Your suggestions have helped us improve the motivation explanation, clarity of the methodology, and the experimental analysis. Below are our responses to the comments you provided.
>
> **W1: The paper’s review of existing methods is overly abstract and insufficiently specific. For example: “On the one hand, label ambiguity and the complex relationships among candidate labels make it difficult for some methods to achieve stable disambiguation in a single pass, leading to limited robustness.” Nearly all PLL methods aim to address precisely this issue, so this statement does not accurately pinpoint the concrete shortcomings of prior work.**
>
> Thank you for the valuable feedback raised by the reviewer. We understand your concern regarding the abstract description of the complex relationship between label ambiguity and candidate labels, and the lack of specific identification of the shortcomings in existing methods.
>
> (1) **Specific Limitations of Existing PLL Methods**
>
> Most existing partial label learning (PLL) methods rely on classifiers that make a single-step prediction to obtain the result. While this approach is effective, it often suffers from instability when dealing with label ambiguity and noise. In the prediction process, the classifier typically relies only on the input features and the candidate label set to make a one-time prediction, which is highly susceptible to label noise and complex relationships, resulting in poor stability. This issue is particularly evident when there is significant label noise or strong label ambiguity, leading to limited robustness.
>
> Specifically, the existing methods face the following problems:
>
> - **Instability in Single-Step Prediction**: Most existing methods use classifiers for one-time predictions, which leads to instability when label ambiguity and noise are complex. Once a prediction error occurs, it is difficult to correct, limiting the accuracy of the results.
> - **Reliance on Single-Step Inference**: Since these methods rely on single-step inference, they cannot gradually adjust and correct errors through multiple predictions and revisions, making them perform poorly on complex datasets, especially in environments with strong label ambiguity.
>
> (2) **Innovation and Advantages of Our Approach**
>
> In contrast to existing methods, our approach adopts a diffusion model, whose core advantage lies in gradually recovering labels through multi-step denoising, rather than making a one-time prediction. The diffusion model progressively denoises and recovers the true labels during the label disambiguation process, providing opportunities for correction at each step. This process not only allows the model to gradually correct errors in the presence of high label uncertainty but also increases the model's stability.
>
> - **Multi-Step Denoising Enhances Stability**: The diffusion model eliminates label noise gradually through a multi-step denoising process. At each step, the model has an opportunity to correct previous mistakes, ultimately achieving more accurate label predictions. This multi-step denoising significantly enhances label disambiguation stability, especially in environments with high label noise.
> - **Error Correction Capability**: Compared to traditional single-step prediction methods, the diffusion model has a built-in error correction capability. During the denoising process, if an error occurs, subsequent steps can still correct it, avoiding the instability and error accumulation issues associated with single-step predictions.
>
> Additionally, our DAFF module combines deep and shallow features from pre-trained and untrained encoders, improving label disambiguation stability by capturing more semantic information. The FLMI mechanism optimizes mutual information between labels and features, improving label prediction accuracy. A label preprocessing mechanism filters noisy labels early on, reducing their impact on training. The dynamic label updating mechanism gradually adjusts labels, ensuring better alignment with image features and enhancing model stability.
>
> By integrating the diffusion model with these modules, our approach significantly improves label disambiguation stability and accuracy, particularly in environments with high label noise and ambiguity. Experimental results demonstrate that our MDMPLL method outperforms existing state-of-the-art PLL methods, offering higher precision and stability, especially in complex scenarios. Additional explanations are provided in the paper.

---

> ### Author Response · Authors · 2025-11-20
>
> **W2 and Q2: Regarding “On the other hand, label disambiguation and feature learning are often decoupled, and lack mechanisms that effectively associate labels with the semantic content of images.” The intended claim seems to be that disambiguation and representation learning are decoupled and lack a mechanism to link them for stable, mutually reinforcing training. However, the proposed DAFF targets only representation learning and does not provide a joint mechanism. Moreover, many existing PLL approaches already consider disambiguation and representation learning jointly rather than as two separate stages. The proposed DAFF module appears to act only on representation learning and does not resolve the decoupling between disambiguation and representation learning found in prior methods.**
>
> Thank you for the valuable comments on our paper. We understand your concerns, especially regarding whether the DAFF and FLMI modules effectively provide a joint mechanism and how they compare with existing PLL methods.
>
> First, we would like to clarify that our approach differs significantly from traditional PLL methods in terms of label disambiguation and feature learning. Many existing PLL methods do consider label disambiguation and feature learning, but these processes are typically carried out asynchronously. For example, Tian et al. (2023) [1] pointed out that most PLL methods still follow a sequential process, where feature learning is followed by label learning or label prediction is followed by feature fine-tuning. Xia et al. (2023) [2] mainly focus on the role of representation learning in label disambiguation, where the process first enhances feature representation quality and then uses it for label disambiguation. Specifically, existing methods often first perform feature learning (e.g., using neural networks to extract features), and then use these features for label prediction through a classifier. This process involves learning features first and then performing label prediction via a classifier, meaning label disambiguation and feature learning are treated separately. Even when performed simultaneously, these processes are often independent and do not interfere with each other.
>
> In contrast, our approach performs label disambiguation and feature learning simultaneously within the noise prediction network. Specifically, our DAFF and FLMI modules are not trained in separate steps, but are synchronized during the training of the entire network. The DAFF module, through the fusion of multi-level features, produces fused features that are integrated with labels and guide the label disambiguation process. The FLMI module maximizes the mutual information between features and labels to ensure that the correlation between the two is optimized. It directly influences feature learning during the label disambiguation process, ensuring that the labels remain consistent with the semantic content of the images, thereby enhancing the label disambiguation effect. Furthermore, our label updating mechanism ensures that label predictions are continuously refined throughout the training process. This mechanism updates the labels at each iteration, correcting any errors that may have occurred earlier, thus achieving synchronized optimization of label disambiguation and feature learning.
>
> Compared to traditional PLL methods, our approach synchronously performs label disambiguation and feature learning within the noise prediction network, avoiding the drawbacks of asynchronous training in conventional methods. Traditional methods treat label disambiguation and feature learning as two independent stages, while our approach integrates the two processes, allowing them to mutually enhance each other. This results in improved model stability and robustness, especially in scenarios with high label noise or strong label ambiguity. We have also provided additional explanations for this section in the paper.

---

> > ### Author Response · Authors · 2025-11-20
> >
> > **W3: The paper does not explain why diffusion models are suitable for label disambiguation in PLL, which is crucial. Since the label information fed into the diffusion model is noisy, if the denoising process outputs a noisy label at each step, can it ultimately recover the true label? This justification is missing.**
> >
> > Thank you for the valuable feedback on our paper. We understand your concerns about how the diffusion model effectively recovers the true labels during the label disambiguation process, especially when noisy labels are input. To address this question, we would like to further clarify the role of the diffusion model in the denoising process and how our approach, through other modules such as DAFF and FLMI, enhances the effectiveness of label disambiguation.
> >
> > We have described in Appendix C why diffusion models can be used for PLL. Additionally, although each step does indeed output a noisy label, the noise in the output label gradually decreases and approaches the correct label. The diffusion model performs label correction step by step through multiple denoising steps. After the initial noisy label goes through the progressive denoising process, the impact of the noise is gradually reduced, ultimately recovering the true label with the highest confidence.
> >
> > In addition to the step-by-step denoising process of the diffusion model, our approach also incorporates the DAFF module and the FLMI mechanism, further enhancing the stability and accuracy of label disambiguation. The DAFF module provides richer feature representations by fusing deep and shallow features. These fused features assist in label prediction during the denoising process, making the denoising of labels not only reliant on the labels themselves but also guided effectively by multi-level image features, thus improving label disambiguation. The FLMI mechanism strengthens the relationship between labels and features by maximizing the mutual information between labels and image features. During the denoising process, the FLMI mechanism ensures that the labels are tightly aligned with the semantic content of the image, making the recovered labels more precise and minimizing the influence of noisy labels.
> >
> > Ultimately, the labels recovered by the diffusion model are not noisy labels but rather labels with high class confidence. Through the multi-step denoising process of the diffusion model and the joint optimization of DAFF and FLMI, the label disambiguation effect is significantly enhanced, and the recovered labels exhibit higher stability and accuracy. The step-by-step denoising mechanism of the diffusion model continuously reduces the noise in the labels, making the final recovered labels closer to the true labels. Meanwhile, the DAFF and FLMI modules further optimize the denoising process by providing stronger feature representations and enhancing the correlation between labels and features, ensuring the accuracy and stability of label recovery.

---

> > > ### Author Response · Authors · 2025-11-20
> > >
> > > **Q1**
> > >
> > > Thank you for the valuable questions raised by the reviewer. Below are my detailed responses.
> > >
> > > **(1) Fundamental Issues in Applying Diffusion Models to PLL**
> > >
> > > Diffusion models typically rely on clean, fully observed images as input for tasks like image denoising. However, in PLL, the input labels are noisy, and there is inherent uncertainty about the true label. Therefore, applying diffusion models to PLL does indeed present a challenge: the denoising process cannot initially rely on clear label information. However, the strength of diffusion models lies in their gradual denoising process. Even if the input labels contain noise, the diffusion model gradually refines the labels over multiple steps, fine-tuning them at each stage, reducing the noise, and ultimately recovering the true label. This step-by-step correction process can effectively mitigate the problem of label noise, making diffusion models highly promising for PLL.
> > >
> > > **(2) Why Use KNN Initialization for Disambiguation**
> > >
> > > In our model, KNN initialization is used to initialize labels by leveraging the similarity between labels and the similarity between features, thereby improving the stability of label disambiguation. The key idea behind KNN initialization is to find samples that are similar to the current sample in label space, using the features and label information of these similar samples to help predict the current sample's label. This approach has several important advantages. It reduces the impact of label noise by helping eliminate some of the incorrect predictions caused by label noise. By using this initialization, the model starts with a more reasonable label, avoiding training from entirely random noisy labels, which enhances the stability of the training process. This method also improves the quality of the initial labels, as labels from similar samples are semantically closer, and KNN initialization can effectively enhance the accuracy of the initial labels. This helps make the denoising process of the diffusion model more efficient, as higher-quality input labels allow for easier recovery of true labels from noisy labels.
> > >
> > > **(3) Why Noisy Labels Can Still Be Used as Input for Diffusion Models**
> > >
> > > Even though the input labels contain noise, the step-by-step denoising process of the diffusion model allows us to recover the true labels from noisy labels during training. In the framework of the diffusion model, each step fine-tunes the labels through interaction with features (via DAFF and FLMI), gradually reducing the noise and bringing the labels closer to the true label. Specifically, even if the labels are noisy at the input, the denoising process of the diffusion model can progressively improve the label predictions with each iteration. This process gradually makes the labels closer to the true labels, while the maximization of mutual information between features and labels ensures that the labels align with the semantic content of the images, thus effectively reducing the impact of noisy labels on the label prediction.
> > >
> > > **(4) How KNN Initialization Improves Disambiguation Stability**
> > >
> > > KNN initialization helps improve the stability of label disambiguation. Specifically, without KNN initialization, the diffusion model would start denoising from random or incorrect labels, which makes the denoising process more difficult and unstable. KNN initialization provides a more reasonable starting point for the labels, allowing the diffusion model’s denoising process to proceed more smoothly. KNN initialization also speeds up the model's convergence: when the initial labels are closer to the true labels, fewer denoising steps are required, accelerating convergence. This is particularly important for high-noise datasets, as it reduces the interference of noise during model training.
> > >
> > > **(5) What Issues Might Arise Without KNN Initialization?**
> > >
> > > For simpler datasets like MNIST, the performance decrease without KNN initialization may be smaller. However, for datasets with higher noise levels or more complexity, such as CIFAR-100, the issues may be more significant. Without KNN initialization, problems such as difficulty in denoising, unstable training, and slower convergence may arise.
> > >
> > > In summary, the application of diffusion models in PLL effectively addresses label noise and ambiguity through the step-by-step denoising mechanism. KNN initialization, as an effective preprocessing method, improves the quality of the initial labels, thus enhancing the stability of the disambiguation process and accelerating model convergence. Without KNN initialization, the model’s training could be severely affected by label noise, leading to an unstable training process and slower convergence. Therefore, KNN initialization is highly effective for our diffusion model in the context of PLL.

---

> > > > ### Author Response · Authors · 2025-11-20
> > > >
> > > > **Q3: From the ablation results, the contributions of components D and F are modest, yet the overall gain over existing methods is large. This suggests the diffusion model itself contributes substantially. It would be better to show ablations across multiple datasets.**
> > > >
> > > > Thank you for the valuable suggestions raised by the reviewer. In response, we conducted ablation experiments on the MSRCv2 and BirdSong datasets. The experimental results have been updated in the paper. From the results, we can clearly observe the improvement that the FLMI and DAFF modules bring to PLL disambiguation. When using each module individually, there is already a significant accuracy improvement compared to the baseline, indicating that each module plays a key role in label disambiguation. When both modules are used together, there is a substantial increase in accuracy, demonstrating the complementary effect of the two modules. Combining both modules results in better overall performance.
> > > >
> > > > | Ablation          | D    | F    | CIFAR-100          | Lost               | MSRCv2             | BirdSong           |
> > > > | ----------------- | ---- | ---- | ------------------ | ------------------ | ------------------ | ------------------ |
> > > > | **MDMPLL**        | ✔    | ✔    | $83.59 \pm 0.17\%$ | $76.79 \pm 0.27\%$ | $56.57 \pm 0.16\%$ | $75.15 \pm 0.13\%$ |
> > > > | **MDMPLL-w/o-D**  | ✘    | ✔    | $82.25 \pm 0.13\%$ | $71.43 \pm 0.32\%$ | $51.43 \pm 0.23\%$ | $71.74 \pm 0.15\%$ |
> > > > | **MDMPLL-w/o-F**  | ✔    | ✘    | $82.84 \pm 0.16\%$ | $74.11 \pm 0.28\%$ | $53.14 \pm 0.09\%$ | $72.95 \pm 0.21\%$ |
> > > > | **MDMPLL-w/o-DF** | ✘    | ✘    | $81.42 \pm 0.21\%$ | $68.75 \pm 0.19\%$ | $50.86 \pm 0.15\%$ | $69.74 \pm 0.26\%$ |
> > > >
> > > > **Q4: After training, can the model use the diffusion denoising process to directly predict labels for unseen examples?**
> > > >
> > > > Thank you for the valuable question raised by the reviewer. Regarding your question, "After training, can the model use the diffusion denoising process to directly predict the labels of unseen samples?" we are happy to provide a detailed explanation.
> > > >
> > > > After training, our method can predict the categories of unseen samples, but the condition is that the category of the sample must have been seen during training. During the training phase, we train a label prediction network, and the DAFF module and FLMI are continuously updated throughout the process, making the label disambiguation more stable. When encountering new samples, we use both the untrained encoder and the pre-trained encoder to extract features from the new sample. These features are then passed through the DAFF module to obtain the fused features, which guide the label recovery. The initial predicted label is noise that follows a Gaussian distribution and is random, but with the guidance of the DAFF fused features, the adjustment of the FLMI mechanism, and the denoising ability of the diffusion model prediction network itself, we can fully recover the true label for that sample. Therefore, our label prediction is not solely based on the denoising process of the diffusion model. The DAFF module and FLMI mechanism also play a crucial role in label disambiguation.

---

> > > > > ### Author Response · Authors · 2025-11-20
> > > > >
> > > > > **Q5: In Eq. (15), what does $V$ denote? What is the relationship between Eq. (9) and Eqs. (10), (15), and (16)?**
> > > > >
> > > > > Thank you for the valuable question raised by the reviewer.
> > > > >
> > > > > In Equation (15), $V$ represents the similarity score between two samples, which is a function of similarity calculation. Here, it refers to the similarity score between the label and its corresponding sample features.
> > > > >
> > > > > Equation (9) represents the variational inference in the diffusion model, aiming to train the model by optimizing the variational lower bound. Equation (10) is the final loss function of the diffusion model. The relationship between the variational lower bound ($L_{VLB}$) and the loss function ($L_d$) is that the loss function ($L_d$) is typically optimized by adjusting certain parts of the variational lower bound ($L_{VLB}$). From the perspective of variational inference,$L_{VLB}$ approximates the true data distribution by maximizing the lower bound. The loss function $L_d$ reflects the noise prediction error at each time step $t$, directly involving the model's reverse process (recovering data from noise). The KL divergence term in $L_{VLB}$ can be seen as a measure of the noise error, as it evaluates the difference between the true posterior distribution and the model's inferred distribution, indirectly influencing the learning of noise. During training, $\boldsymbol{\epsilon}_{\theta}$ approximates the true noise (i.e., the noise of the original data) by minimizing $L_d$. Therefore, in the diffusion model, $L_d$ is a final expression for optimizing the variational lower bound.
> > > > >
> > > > > Equation (15) represents the mutual information loss. Equation (16) is the total loss of the entire model, which includes the mutual information loss in Equation (15) and the diffusion model loss in Equation (9).
> > > > >
> > > > > [1] Tian, Yingjie, et al. "Partial label learning: Taxonomy, analysis and outlook." *Neural Networks* 161 (2023): 708-734.
> > > > >
> > > > > [2] Xia, Shiyu, et al. "Towards effective visual representations for partial-label learning." *Proceedings of the IEEE/CVF Conference on Computer Vision and Pattern Recognition* (2023): 15589-15598.

---

### Official Review · Reviewer_dPb4 · 2025-10-31

**Soundness:** 3
**Presentation:** 3
**Contribution:** 2
**Rating:** 4
**Confidence:** 3

**Summary:**

This paper proposes MDMPLL, a diffusion-based framework for partial label learning (PLL).
The authors treat label ambiguity as a forward diffusion process that gradually injects noise into labels and design a reverse denoising process for label disambiguation.
They further incorporate a Dual-Path Attention Feature Fusion (DAFF) module to combine shallow and deep features, and a Feature–Label Mutual Information (FLMI) objective to encourage alignment between features and labels.
Experimental results on several benchmark PLL datasets demonstrate that the proposed method performs competitively compared to prior approaches.

**Strengths:**

1.	The attempt to interpret label ambiguity via diffusion dynamics is novel and interesting.
	2.	The idea of integrating mutual information maximization into label disambiguation is conceptually meaningful.
	3.	The paper is generally well written and easy to follow.

**Weaknesses:**

1. Although Section 3 provides mathematical formulations, the approach largely mirrors standard DDPM procedures with only superficial adaptation to labels. The claimed connection between diffusion and mutual information is not derived or justified; no analysis is provided on how FLMI quantitatively improves disambiguation or generalization.
2. No ablation on the diffusion depth or comparison with simpler PLL baselines using consistency regularization is provided.
3. Visualization results (Fig. 4–5) are qualitative and do not convincingly demonstrate the claimed progressive disambiguation.

**Questions:**

See weaknesses.

---

> ### Author Response · Authors · 2025-11-20
>
> Dear Reviewer dPb4,
>
> Thank you for your thorough review of our paper and your valuable feedback. Your suggestions have helped us improve the explanation of the motivation, the clarity of the method, and the experimental analysis. Below is our response to the comments you raised.
>
> **W1: Although Section 3 provides mathematical formulations, the approach largely mirrors standard DDPM procedures with only superficial adaptation to labels. The claimed connection between diffusion and mutual information is not derived or justified; no analysis is provided on how FLMI quantitatively improves disambiguation or generalization.**
>
> Thank you for the reviewer’s valuable comments. We understand your concern regarding theoretical rigor, but we would like to clarify that the core contribution of this paper is not the proposal of a completely new diffusion mechanism or mutual information estimation theory, but rather the synergistic application of diffusion models and mutual information maximization in the specific context of PLL, and the effective label disambiguation achieved through architectural design.
>
> Although the forward/backward process borrows from DDPM, the target and objectives are entirely different. DDPM typically operates in pixel space to generate images, while we simulate noise injection and denoising in the candidate label space, aiming to recover the unique true label from multiple candidate labels. This transition from "generative modeling" to "discriminative label purification" is non-trivial and introduces new challenges, such as label discreteness and candidate set constraints. Our framework substantially adapts DDPM through the introduction of a candidate mask mechanism and a reverse process guided by semantic features, rather than simply applying it.
>
> Our diffusion mechanism is responsible for dynamically optimizing pseudo-labels, while the FLMI loss serves as an independent but complementary supervisory signal to enhance the consistency between features and labels. The two modules are complementary: the diffusion process provides a temporal refinement path for label evolution, while FLMI offers discriminative constraints at each step.
>
> Additionally, the discussion on how mutual information maximization facilitates label disambiguation can be found in Appendix E. In our experiments, we have verified the role of FLMI through ablation studies. To further highlight the importance of the FLMI and DAFF modules, we conducted additional ablation experiments on the MSRCv2 and BirdSong datasets, which also confirmed the significance of the FLMI and DAFF modules.
>
> FLMI maximizes the mutual information between a label and its corresponding feature, while minimizing the mutual information between the label and unrelated features. The features thus contain sufficient information about the category, which allows the true label to emerge from the candidate labels. Moreover, the features are also closer to the true label, and features from the same category become even closer, further aiding label disambiguation.
>
> | Ablation          | D    | F    | CIFAR-100          | Lost               | MSRCv2             | BirdSong           |
> | ----------------- | ---- | ---- | ------------------ | ------------------ | ------------------ | ------------------ |
> | **MDMPLL**        | ✔    | ✔    | $83.59 \pm 0.17\%$ | $76.79 \pm 0.27\%$ | $56.57 \pm 0.16\%$ | $75.15 \pm 0.13\%$ |
> | **MDMPLL-w/o-D**  | ✘    | ✔    | $82.25 \pm 0.13\%$ | $71.43 \pm 0.32\%$ | $51.43 \pm 0.23\%$ | $71.74 \pm 0.15\%$ |
> | **MDMPLL-w/o-F**  | ✔    | ✘    | $82.84 \pm 0.16\%$ | $74.11 \pm 0.28\%$ | $53.14 \pm 0.09\%$ | $72.95 \pm 0.21\%$ |
> | **MDMPLL-w/o-DF** | ✘    | ✘    | $81.42 \pm 0.21\%$ | $68.75 \pm 0.19\%$ | $50.86 \pm 0.15\%$ | $69.74 \pm 0.26\%$ |

---

> > ### Author Response · Authors · 2025-11-20
> >
> > **W2: No ablation on the diffusion depth or comparison with simpler PLL baselines using consistency regularization is provided.**
> >
> > Thank you for the reviewer’s valuable comments. First, we conducted hyperparameter experiments on the number of diffusion steps $t$, and the experimental results show that the best performance is achieved when $t = 1000$.
> >
> > Next, we also included two comparison experiments: CRDPLL and PICO. CRDPLL is a typical PLL method that enhances model performance through consistency regularization. Although PICO is not a traditional consistency regularization method, it shares a similar idea. It works by pulling similar samples closer together and pushing dissimilar samples apart, thereby learning data representations to achieve label disambiguation.
> >
> > The specific experimental results have been included in the paper.
> >
> > **W3: Visualization results (Fig. 4–5) are qualitative and d not convincingly demonstrate the claimed progressive disambiguation.**
> >
> > Thank you for the feedback raised by the reviewer. Figure 5 shows the visualization of the final prediction results, and it indeed does not cover the progressive label disambiguation process. However, in Figure 4, we do show the process of progressive label disambiguation. This figure illustrates the changes in clustering after each diffusion step during training, reflecting the model's gradual improvement in label disambiguation. Each round of the diffusion steps helps the model progressively refine noisy labels into more accurate label predictions. Therefore, the design of Figure 4 itself already captures the progressive nature of label disambiguation, with the label changes at each diffusion step reflecting how the model gradually removes label noise and improves prediction accuracy.

---

> > > ### Author Response · Authors · 2025-11-20
> > >
> > > | Method              | $q$  | MNIST                       | Fashion-MNIST               | Kuzushiji-MNIST             | CIFAR-10                    | $q$  | CIFAR-100                       |
> > > | ------------------- | ---- | --------------------------- | --------------------------- | --------------------------- | --------------------------- | ---- | ------------------------------- |
> > > |                     | 0.1  | $98.97 \pm 0.12\%$          | $\mathbf{93.36 \pm 0.09\%}$ | $\mathbf{97.68 \pm 0.06\%}$ | $94.54 \pm 0.05\%$          | 0.01 | $71.02 \pm 0.29\%$              |
> > > | **PiCO**            | 0.3  | $98.85 \pm 0.14\%$          | $\mathbf{93.12 \pm 0.12\%}$ | $\mathbf{97.34 \pm 0.03\%}$ | $94.13 \pm 0.08\%$          | 0.05 | $70.29 \pm 0.29\%$              |
> > > |                     | 0.5  | $98.63 \pm 0.03\%$          | $\mathbf{92.88 \pm 0.03\%}$ | $\mathbf{97.15 \pm 0.12\%}$ | $93.85 \pm 0.15\%$          | 0.1  | $58.16 \pm 0.34\%$              |
> > > |                     | 0.1  | $98.72 \pm 0.22\%$          | $93.21 \pm 0.24\%$          | $97.13 \pm 0.57\%$          | $94.71 \pm 0.18\%$          | 0.01 | $75.31 \pm 0.61\%$              |
> > > | **CRDPLL**          | 0.3  | $98.64 \pm 0.19\%$          | $92.53 \pm 0.31\%$          | $96.55 \pm 0.45\%$          | $94.27 \pm 0.24\%$          | 0.05 | $76.67 \pm 0.18\%$              |
> > > |                     | 0.5  | $98.32 \pm 0.33\%$          | $91.47 \pm 0.20\%$          | $96.21 \pm 0.61\%$          | $93.84 \pm 0.11\%$          | 0.1  | $71.21 \pm 0.34\%$              |
> > > |                     | 0.1  | $98.19 \pm 0.11\%$          | $89.96 \pm 0.09\%$          | $95.22 \pm 0.27\%$          | $90.47 \pm 0.13\%$          | 0.01 | $57.53 \pm 0.23\%$              |
> > > | **LRA-Diffusion**   | 0.3  | $97.83 \pm 0.25\%$          | $89.32 \pm 0.24\%$          | $94.73 \pm 0.34\%$          | $89.94 \pm 0.29\%$          | 0.05 | $55.22 \pm 0.23\%$              |
> > > |                     | 0.5  | $97.59 \pm 0.13\%$          | $86.38 \pm 0.10\%$          | $92.72 \pm 0.16\%$          | $86.86 \pm 0.34\%$          | 0.1  | $49.48 \pm 0.09\%$              |
> > > |                     | 0.1  | $98.39 \pm 0.10\%$          | $89.75 \pm 0.12\%$          | $96.01 \pm 0.12\%$          | $90.71 \pm 0.12\%$          | 0.01 | $66.50 \pm 0.18\%$              |
> > > | **MDMPLL (SimCLR)** | 0.3  | $98.27 \pm 0.12\%$          | $89.28 \pm 0.14\%$          | $95.77 \pm 0.06\%$          | $90.47 \pm 0.06\%$          | 0.05 | $65.92 \pm 0.14\%$              |
> > > |                     | 0.5  | $98.06 \pm 0.03\%$          | $88.49 \pm 0.08\%$          | $94.92 \pm 0.11\%$          | $89.74 \pm 0.05\%$          | 0.1  | $64.65 \pm 0.07\%$              |
> > > |                     | 0.1  | $\mathbf{99.19 \pm 0.06\%}$ | $92.00 \pm 0.12\%$          | $92.60 \pm 0.05\%$          | $\mathbf{97.71 \pm 0.11\%}$ | 0.01 | **$\mathbf{83.95 \pm 0.37\%}$** |
> > > | **MDMPLL (CLIP)**   | 0.3  | $\mathbf{99.00 \pm 0.15\%}$ | $91.43 \pm 0.09\%$          | $91.04 \pm 0.14\%$          | $\mathbf{97.57 \pm 0.15\%}$ | 0.05 | **$\mathbf{83.59 \pm 0.17\%}$** |
> > > |                     | 0.5  | $\mathbf{98.77 \pm 0.11\%}$ | $90.25 \pm 0.12\%$          | $90.44 \pm 0.12\%$          | $\mathbf{97.35 \pm 0.23\%}$ | 0.1  | $\mathbf{83.63 \pm 0.03\%}$     |
> > >
> > > | Method            | Lost                        | MSRCv2                      | Birdsong                    | SoccerPlayer                | YahooNews                   |
> > > | ----------------- | --------------------------- | --------------------------- | --------------------------- | --------------------------- | --------------------------- |
> > > | **PiCO**          | $65.33 \pm 0.75\%$          | $49.14 \pm 0.57\%$          | $61.29 \pm 2.10\%$          | $55.13 \pm 1.48\%$          | $\mathbf{68.71 \pm 0.22\%}$ |
> > > | **CRDPLL**        | $64.55 \pm 0.31\%$          | $49.14 \pm 0.87\%$          | $72.00 \pm 0.62\%$          | $54.47 \pm 0.22\%$          | $65.23 \pm 0.74\%$          |
> > > | **LRA-Diffusion** | $65.18 \pm 0.24\%$          | $46.86 \pm 0.13\%$          | $63.53 \pm 0.42\%$          | $50.66 \pm 0.27\%$          | $44.45 \pm 0.34\%$          |
> > > | **MDMPLL**        | $\mathbf{76.79 \pm 0.27\%}$ | $\mathbf{56.57 \pm 0.16\%}$ | $\mathbf{75.15 \pm 0.13\%}$ | $\mathbf{60.26 \pm 0.14\%}$ | $50.66 \pm 0.26\%$          |

---

### Official Review · Reviewer_2MF9 · 2025-11-01

**Soundness:** 3
**Presentation:** 2
**Contribution:** 2
**Rating:** 4
**Confidence:** 4

**Summary:**

This paper introduces MDMPLL, a diffusion-based framework for partial label learning (PLL) that models label ambiguity via forward noise addition to simulate candidate labels and reverse denoising for disambiguation. It incorporates Dual-Path Attention Feature Fusion (DAFF) to blend shallow and deep features as conditional inputs, and maximizes Feature-Label Mutual Information (FLMI) to align labels with semantics. A KNN-based initial disambiguation and dynamic label updates enhance stability.

**Strengths:**

1. nnovative adaptation of diffusion models to PLL, reframing disambiguation as probabilistic denoising with theoretical ties to variational bounds.

2. DAFF effectively fuses multi-level features via attention, improving representation quality; FLMI provides additional supervision for robustness.

3. Comprehensive ablations on components (DAFF, FLMI, diffusion steps) and hyperparameters; strong gains over baselines like PRODEN, CC, RC (e.g., +5-10% on CIFAR-100).

**Weaknesses:**

1. Role of mutual information estimation in disambiguation using diffusion model is not clearly explained; DAFF module seems disconnected from other parts of the method, and its relationship with other modules is unclear.

2. Mutual information guidance, as one of the core contributions, should not be excluded from Figure 1's overall framework.

3. In Algorithm 1 table, isn't the ultimate goal of the algorithm to get a model with strong generalization ability? Why return a clean label matrix?

**Questions:**

1. How does FLMI estimation interact with diffusion variance schedules? Could adaptive scheduling based on MI improve convergence?

2. Why not integrate DAFF into the noise prediction network end-to-end instead of as a separate fusion step?.

---

> ### Author Response · Authors · 2025-11-20
>
> Dear Reviewer 2MF9,
>
> Thank you for your thorough review of our paper and your valuable feedback. Your suggestions have helped us improve the motivation explanation, clarity of the method, and experimental analysis. Below is our response to the comments you raised.
>
> **W1: Role of mutual information estimation in disambiguation using diffusion model is not clearly explained; DAFF module seems disconnected from other parts of the method, and its relationship with other modules is unclear.**
>
> Thank you for the valuable comments raised by the reviewer. I will further clarify the role of mutual information estimation in label disambiguation and the relationship between the DAFF module and other components.
>
> (1) **The Role of Mutual Information Estimation in Label Disambiguation**
>
> In PLL, label disambiguation is a core challenge, as each sample has multiple candidate labels, but only one is correct. Our approach addresses this issue by introducing the FLMI mechanism. The goal of FLMI is to make the matching between labels and features more accurate by estimating the mutual information between them. Specifically, the FLMI mechanism encourages the model to maximize the mutual information between a label and its corresponding feature during training while minimizing the mutual information between the label and other irrelevant features, thus refining the label disambiguation process. This process helps the model select the correct label from multiple candidates and avoid mismatches. By maximizing mutual information, the two variables are brought closer together. In our method, these two variables are the label and its corresponding feature, where the feature is the one fused by the DAFF module, combining shallow visual features and deep semantic features. This feature contains sufficient information about the sample's category, making the correct label among the candidates more apparent. Moreover, the feature becomes closer to the true label, and features of the same class will cluster together, further assisting in label disambiguation.
>
> (2) **The Role of the DAFF Module and Its Relationship with Other Components**
>
> The DAFF module is one of the key parts of our method. It enhances the model's adaptability to the PLL task by fusing deep and shallow features. Specifically, the DAFF module combines deep semantic features from the pre-trained encoder and shallow visual features from the untrained encoder, which are fused using an attention mechanism. This fusion allows the model to utilize both shallow and deep features during label disambiguation, thereby improving the accuracy of label prediction. As mentioned in (1), the DAFF module is closely related to the FLMI mechanism. The features obtained through the DAFF module contain sufficient information about the sample's category, enabling the FLMI to play a more significant role. Meanwhile, FLMI strengthens the mutual information between the label and feature, providing a more stable objective function for the DAFF module, allowing the feature fusion process to better align with label information. This relationship is validated in our experiments, demonstrating that the feature fusion in the DAFF module and the mutual information maximization in FLMI complement each other, jointly enhancing the label disambiguation effect.
>
> Therefore, the DAFF module is not disconnected from the other components but is tightly linked to the FLMI and label disambiguation process. Through feature fusion and mutual information maximization, our model achieves higher accuracy and more stable training in PLL tasks. We have made additions in both the Introduction and Ablation Study sections of the paper.

---

> > ### Author Response · Authors · 2025-11-20
> >
> > **W2: Mutual information guidance, as one of the core contributions, should not be excluded from Figure 1's overall framework.**
> >
> > Thank you for the valuable suggestions raised by the reviewer. We understand that mutual information guidance, as one of the core contributions, should be clearly presented within the method framework. However, regarding the design of Figures 1 and 2, we believe that FLMI should appear in Figure 2 rather than Figure 1, for the following reasons:
> >
> > Figure 1 presents the overall framework of the method, focusing on the forward and backward propagation of the diffusion model, label preprocessing, and the label updating process. It serves as a high-level overview, aiming to provide readers with a general view of the method. Therefore, Figure 1 emphasizes the structural components of the method, without delving into the specific tasks executed by each module.
> >
> > In contrast, Figure 2 illustrates the most critical part of the diffusion model: the noise prediction network, which is the core architecture of the diffusion model. It is through the noise prediction network that we are able to gradually recover the true labels. Since our FLMI, DAFF, and loss functions are closely related to the noise prediction network, we integrated these details together. FLMI, as a key mechanism, directly aligns the relationship between features and labels, making it more suitable for display in Figure 2. In Figure 2, we show how FLMI maximizes the mutual information between labels and features, and how it interacts with DAFF to improve label disambiguation. This is one of the reasons why we decided to place FLMI in Figure 2.
> >
> > The core role of FLMI in the method is to optimize label prediction by estimating the mutual information between features and labels, ensuring their alignment. Since this mechanism operates at a finer level of detail by using the similarity between features and labels, it should be presented in a more detailed diagram to highlight its specific process and interaction with other modules.
> >
> > **W3: In Algorithm 1 table, isn't the ultimate goal of the algorithm to get a model with strong generalization ability? Why return a clean label matrix?**
> >
> > Thank you for the valuable suggestions raised by the reviewer. Our intention was actually that the algorithm not only results in a model with strong generalization ability but also continuously updates the candidate labels to obtain a clean label matrix. We wanted to express the effectiveness of label updates in the algorithm, which is why we included the final step of returning the clean label matrix. We are happy to remove the last line, as it does not affect the meaning we intended to convey.

---

> > > ### Author Response · Authors · 2025-11-20
> > >
> > > **Q1: How does FLMI estimation interact with diffusion variance schedules? Could adaptive scheduling based on MI improve convergence?**
> > >
> > > Thank you for the valuable questions raised by the reviewer. Below is our detailed response to this question:
> > >
> > > (1) **Interaction Between FLMI Estimation and Diffusion Variance Scheduling**
> > >
> > > In our framework, the FLMI mechanism is closely integrated with the diffusion process. The main purpose of FLMI is to maximize the mutual information between labels and features, allowing the model to accurately align labels and features during training, thereby improving the label disambiguation problem. In the diffusion model, variance scheduling controls the intensity of noise addition and denoising. The goal of the diffusion model is to gradually "denoise" over multiple iterations, thereby making the association between labels and features clearer. Specifically, the diffusion variance scheduling determines the noise intensity at each step, while FLMI estimation adjusts the label prediction at each iteration by maximizing the mutual information between labels and features. This means that FLMI helps the model focus on the correct label during each iteration and guides the noise removal process through variance scheduling. In the earlier diffusion steps, when the noise is high, the FLMI mechanism helps the model make effective label inferences even in a more ambiguous label environment. In the later stages of the diffusion process, as the noise decreases, the FLMI mechanism makes the precise alignment of labels more stable.
> > >
> > > (2) **Improvement of Convergence through Mutual Information-Based Adaptive Scheduling**
> > >
> > > The mutual information-based adaptive scheduling has the potential to further improve the model’s convergence. Typically, in standard diffusion models, variance scheduling is predefined, and a fixed scheduling strategy may not fully adapt to the learning process at different stages. The mutual information-based adaptive scheduling can dynamically adjust the noise addition and denoising process based on the current mutual information between labels and features. Specifically, when the FLMI estimation shows a high correlation between labels and features, the noise intensity in the diffusion process will decrease, encouraging the model to denoise and update the labels more precisely. Conversely, when FLMI shows a low similarity between labels and features, the noise intensity will increase, thus preventing the model from making incorrect label predictions too early. Therefore, FLMI-based adaptive scheduling can make the model more flexible and efficient in converging during different training stages, especially when dealing with datasets with high label ambiguity, which can accelerate convergence and improve final performance.
> > >
> > > FLMI estimation works directly with the noise removal mechanism in the diffusion process by maximizing the mutual information between labels and features. The mutual information-based adaptive scheduling can dynamically adjust the noise intensity based on the alignment between labels and features, potentially improving the convergence of the diffusion model and enhancing the final label accuracy and generalization ability.

---

> > > > ### Author Response · Authors · 2025-11-20
> > > >
> > > > **Q2: Why not integrate DAFF into the noise prediction network end-to-end instead of as a separate fusion step?.**
> > > >
> > > > Thank you for the valuable question raised by the reviewer. In our training process, the DAFF module is indeed integrated into the noise prediction network as an essential component of the noise prediction process. We chose to explain the DAFF module separately rather than merging it with the noise prediction network for the following reasons:
> > > >
> > > > (1) **DAFF as a Key Component of the Noise Prediction Network**: The DAFF module is actually a core step within the noise prediction network, focusing on the fusion of deep and shallow features. In PLL tasks, feature diversity is crucial, and the DAFF module enhances the noise prediction network’s ability to model noise by combining information from the pre-trained encoder (deep features) and the untrained encoder (shallow features). This fusion efficiently captures features at different levels through an attention mechanism, making the noise prediction process more accurate.
> > > >
> > > > (2) **Why DAFF is Explained Separately**: Although the DAFF module is embedded in the noise prediction network, it performs a very important and complex task. We chose to explain it separately not only to clearly showcase its function but also to highlight its relationship with the noise prediction process. By effectively fusing features at different levels, the DAFF module helps the noise prediction network maintain high stability and accuracy in the face of label ambiguity.
> > > >
> > > > (3) **The Relationship Between DAFF and FLMI**: The DAFF module is closely related to the FLMI mechanism. The goal of FLMI is to strengthen the correlation between labels and features, ensuring that the model can accurately predict labels and reduce ambiguity. In the noise prediction network, the DAFF module ensures the effective flow of information by fusing deep and shallow features. The FLMI mechanism further helps by optimizing the mutual information between labels and features, enabling the model to continually adjust label predictions during training. Therefore, the collaborative work of DAFF and FLMI makes the label disambiguation process more refined and stable.
> > > >
> > > > (4) **Advantages of Modular Design**: Although DAFF is part of the noise prediction network, due to its central role in noise modeling and label disambiguation, explaining it separately helps readers better understand its role in the model. By distinguishing DAFF from the other parts of the noise prediction network, we can more effectively demonstrate its unique contributions in feature fusion, label alignment, and noise removal.
> > > >
> > > > The DAFF module is indeed a part of the noise prediction network, but because of its key role in the label disambiguation process, we chose to explain it as an important component separately, emphasizing its relationship with FLMI. By doing so, we are able to clearly show the importance of the DAFF module within the noise prediction network and its synergistic effect with the mutual information maximization mechanism, further enhancing the model's generalization ability.

---

### Official Review · Reviewer_jV29 · 2025-11-02

**Soundness:** 2
**Presentation:** 2
**Contribution:** 2
**Rating:** 4
**Confidence:** 5

**Summary:**

The work presents a method for partial label learning or label disambiguation by invoking a diffusion process in the label space and progressively noising and denoising the labels to learn a robust denoiser. The method is guided by mutual information between labels and features derived from the shallow and deep sections of encoders, fused using Dual Path Attention Feature Fusion, eventually maximizing Feature Label Mutual Information using the InfoNCE loss.

**Strengths:**

1. The method shows decent empirical gains against baselines.
2. Incorporation of attention based features and MI guidance may be well informed in this problem and leads to demonstrable gains.

**Weaknesses:**

1. Some of the proofs are hand-wavy and/or already shown (Appendix D and E). The forward and reverse processes come from diffusion models literature. Appendix F outlines known results from EM literature. These proofs don't incorporate the specifics of the method proposed to synthesize something novel.

2. Comparisons against LRA-Diffusion [1] are missing which also uses pre-trained network features as guidance.

3. The authors could shed more light into the rationale behind using both pre-trained and un-trained encoders for deep and shallow feature sets respectively.

4. The motivation behind label and feature vector similarity may be mis-guided. Labels lie in a categorical simplex, whereas features are continuous embeddings. The authors could justify this choice in principle.

5. Training and inference runtime overheads for the methods aren't reported.


[1] Chen, Jian, et al. "Label-retrieval-augmented diffusion models for learning from noisy labels." Advances in Neural Information Processing Systems 36 (2023): 66499-66517.

**Questions:**

1. Is there a study that demonstrates the effects of the label filtering stage at the beginning?
2. Could the authors kindly delineate the differences between LRA-Diffusion and this work?

[1] Chen, Jian, et al. "Label-retrieval-augmented diffusion models for learning from noisy labels." Advances in Neural Information Processing Systems 36 (2023): 66499-66517.

---

> ### Author Response · Authors · 2025-11-20
>
> Dear Reviewer jV29,
>
> Thank you for your thorough review of our paper and your valuable feedback. Your suggestions have helped us improve the motivation explanation, clarity of the method, and experimental analysis. Below is our response to the comments you raised.
>
> **W1:Some of the proofs are hand-wavy and/or already shown (Appendix D and E). The forward and reverse processes come from diffusion models literature. Appendix F outlines known results from EM literature. These proofs don't incorporate the specifics of the method proposed to synthesize something novel.**
>
> Thank you for the questions raised by the reviewer. Appendix D discusses how the DAFF module contributes to more accurate mutual information estimation. Its core idea is based on the concepts of sufficient statistics and Markov chains. The goal is to provide an information-theoretic rationale for the design of the DAFF module, explaining why the fused features better retain information related to the labels, thereby improving the effectiveness of FLMI. This proof uses classical information-theoretic results to explain the module's effectiveness.
>
> Appendix E shows that the FLMI loss is essentially the negative InfoNCE loss, and by leveraging the lower bound properties of InfoNCE on mutual information, it explains that minimizing FLMI reduces the conditional entropy $H(\mathbf{Y}|\mathbf{x})$, forcing the model's predictive distribution  $p_θ(\mathbf{y}∣\mathbf{x})$ to concentrate on the unique correct label within the candidate label set, thus achieving label disambiguation. The purpose of Appendix E is to establish the connection between the loss function and mutual information maximization, and to explain why this loss effectively addresses the noise issue in PLL.
>
> Appendix F derives the label update mechanism from the EM perspective, aiming to provide an explanation of the iterative optimization strategy within the framework of classical statistical learning, ensuring the stability of the training process by utilizing the monotonic convergence property of the EM algorithm. Therefore, Appendix F does not propose a new EM algorithm but rather shows that the proposed update rule aligns with the general form of the EM algorithm, inheriting its desirable convergence properties.
>
> In summary, the content in the appendices is not entirely new mathematical proof but intentionally draws on and references existing theoretical results to demonstrate the effectiveness of the DAFF design, explain the motivation behind the loss function, and provide theoretical support for the optimization process. The core purpose of the appendices is to serve as a bridge connecting mature theories with the proposed method, ensuring that the MDMPLL framework has a clear theoretical foundation for partial label learning tasks.

---

> > ### Author Response · Authors · 2025-11-20
> >
> > **W2 and Q2: Comparisons against LRA-Diffusion [1] are missing which also uses pre-trained network features as guidance.  Could the authors kindly delineate the differences between LRA-Diffusion and this work?**
> >
> > Thank you for the reviewer's suggestion. LRA-Diffusion and our work differ significantly in terms of methodology and objectives. LRA-Diffusion primarily focuses on the issue of noisy labels, where each sample has one correct label but may contain incorrect labels due to label noise (e.g., label flipping). However, in partial label learning (PLL), each sample may have multiple candidate labels, with only one being the correct label. The correct label is unknown during training, making label ambiguity the core challenge of PLL.
> >
> > LRA-Diffusion addresses noise by processing known labels but cannot directly handle cases with multiple candidate labels. The performance of LRA-Diffusion may degrade sharply in cases of high noise because it depends on label clarity. In contrast, our approach models the similarity between labels and features and combines dynamic label and feature update mechanisms to more effectively disambiguate labels. Specifically, we construct a KNN based on the similarity between features and labels and update label predictions through backpropagation at each training step to gradually improve label accuracy.
> >
> > Additionally, LRA-Diffusion guides label prediction by simply concatenating features extracted by a pre-trained encoder with labels, whereas our method uses the DAFF module to attentively fuse shallow and deep features extracted from both pre-trained and untrained encoders. This fusion strategy allows the model to simultaneously leverage shallow visual features and deep semantic features, resulting in richer representation capabilities. Furthermore, by introducing the FLMI mechanism, we further enhance the correlation between labels and features, making our model more adaptable and robust when handling partial label learning tasks.
> >
> > Finally, we attempted to run LRA-Diffusion on the CIFAR-100 dataset with a label noise rate of $0.1$, and we found that the final result was only $49.48%$, which is significantly lower than our result. This further confirms that LRA-Diffusion is not suitable for our partial label learning problem.
> >
> > In summary, the core advantage of LRA-Diffusion lies in handling noisy labels, but its approach faces challenges when applied to PLL problems. Our method, by combining label and feature similarity, mutual information maximization, and feature fusion strategies, presents an innovative solution tailored for partial label learning.

---

> > > ### Author Response · Authors · 2025-11-20
> > >
> > > | Method              | $q$  | MNIST                       | Fashion-MNIST               | Kuzushiji-MNIST             | CIFAR-10                    | $q$  | CIFAR-100                       |
> > > | ------------------- | ---- | --------------------------- | --------------------------- | --------------------------- | --------------------------- | ---- | ------------------------------- |
> > > |                     | 0.1  | $98.97 \pm 0.12\%$          | $\mathbf{93.36 \pm 0.09\%}$ | $\mathbf{97.68 \pm 0.06\%}$ | $94.54 \pm 0.05\%$          | 0.01 | $71.02 \pm 0.29\%$              |
> > > | **PiCO**            | 0.3  | $98.85 \pm 0.14\%$          | $\mathbf{93.12 \pm 0.12\%}$ | $\mathbf{97.34 \pm 0.03\%}$ | $94.13 \pm 0.08\%$          | 0.05 | $70.29 \pm 0.29\%$              |
> > > |                     | 0.5  | $98.63 \pm 0.03\%$          | $\mathbf{92.88 \pm 0.03\%}$ | $\mathbf{97.15 \pm 0.12\%}$ | $93.85 \pm 0.15\%$          | 0.1  | $58.16 \pm 0.34\%$              |
> > > |                     | 0.1  | $98.72 \pm 0.22\%$          | $93.21 \pm 0.24\%$          | $97.13 \pm 0.57\%$          | $94.71 \pm 0.18\%$          | 0.01 | $75.31 \pm 0.61\%$              |
> > > | **CRDPLL**          | 0.3  | $98.64 \pm 0.19\%$          | $92.53 \pm 0.31\%$          | $96.55 \pm 0.45\%$          | $94.27 \pm 0.24\%$          | 0.05 | $76.67 \pm 0.18\%$              |
> > > |                     | 0.5  | $98.32 \pm 0.33\%$          | $91.47 \pm 0.20\%$          | $96.21 \pm 0.61\%$          | $93.84 \pm 0.11\%$          | 0.1  | $71.21 \pm 0.34\%$              |
> > > |                     | 0.1  | $98.19 \pm 0.11\%$          | $89.96 \pm 0.09\%$          | $95.22 \pm 0.27\%$          | $90.47 \pm 0.13\%$          | 0.01 | $57.53 \pm 0.23\%$              |
> > > | **LRA-Diffusion**   | 0.3  | $97.83 \pm 0.25\%$          | $89.32 \pm 0.24\%$          | $94.73 \pm 0.34\%$          | $89.94 \pm 0.29\%$          | 0.05 | $55.22 \pm 0.23\%$              |
> > > |                     | 0.5  | $97.59 \pm 0.13\%$          | $86.38 \pm 0.10\%$          | $92.72 \pm 0.16\%$          | $86.86 \pm 0.34\%$          | 0.1  | $49.48 \pm 0.09\%$              |
> > > |                     | 0.1  | $98.39 \pm 0.10\%$          | $89.75 \pm 0.12\%$          | $96.01 \pm 0.12\%$          | $90.71 \pm 0.12\%$          | 0.01 | $66.50 \pm 0.18\%$              |
> > > | **MDMPLL (SimCLR)** | 0.3  | $98.27 \pm 0.12\%$          | $89.28 \pm 0.14\%$          | $95.77 \pm 0.06\%$          | $90.47 \pm 0.06\%$          | 0.05 | $65.92 \pm 0.14\%$              |
> > > |                     | 0.5  | $98.06 \pm 0.03\%$          | $88.49 \pm 0.08\%$          | $94.92 \pm 0.11\%$          | $89.74 \pm 0.05\%$          | 0.1  | $64.65 \pm 0.07\%$              |
> > > |                     | 0.1  | $\mathbf{99.19 \pm 0.06\%}$ | $92.00 \pm 0.12\%$          | $92.60 \pm 0.05\%$          | $\mathbf{97.71 \pm 0.11\%}$ | 0.01 | **$\mathbf{83.95 \pm 0.37\%}$** |
> > > | **MDMPLL (CLIP)**   | 0.3  | $\mathbf{99.00 \pm 0.15\%}$ | $91.43 \pm 0.09\%$          | $91.04 \pm 0.14\%$          | $\mathbf{97.57 \pm 0.15\%}$ | 0.05 | **$\mathbf{83.59 \pm 0.17\%}$** |
> > > |                     | 0.5  | $\mathbf{98.77 \pm 0.11\%}$ | $90.25 \pm 0.12\%$          | $90.44 \pm 0.12\%$          | $\mathbf{97.35 \pm 0.23\%}$ | 0.1  | $\mathbf{83.63 \pm 0.03\%}$     |
> > >
> > > |      |
> > > | ---- |
> > > |      |
> > >
> > > | Method            | Lost                        | MSRCv2                      | Birdsong                    | SoccerPlayer                | YahooNews                   |
> > > | ----------------- | --------------------------- | --------------------------- | --------------------------- | --------------------------- | --------------------------- |
> > > | **PiCO**          | $65.33 \pm 0.75\%$          | $49.14 \pm 0.57\%$          | $61.29 \pm 2.10\%$          | $55.13 \pm 1.48\%$          | $\mathbf{68.71 \pm 0.22\%}$ |
> > > | **CRDPLL**        | $64.55 \pm 0.31\%$          | $49.14 \pm 0.87\%$          | $72.00 \pm 0.62\%$          | $54.47 \pm 0.22\%$          | $65.23 \pm 0.74\%$          |
> > > | **LRA-Diffusion** | $65.18 \pm 0.24\%$          | $46.86 \pm 0.13\%$          | $63.53 \pm 0.42\%$          | $50.66 \pm 0.27\%$          | $44.45 \pm 0.34\%$          |
> > > | **MDMPLL**        | $\mathbf{76.79 \pm 0.27\%}$ | $\mathbf{56.57 \pm 0.16\%}$ | $\mathbf{75.15 \pm 0.13\%}$ | $\mathbf{60.26 \pm 0.14\%}$ | $50.66 \pm 0.26\%$          |

---

> > > > ### Author Response · Authors · 2025-11-20
> > > >
> > > > **W3**
> > > >
> > > > Thank you for the reviewer’s suggestion. We chose to use pre-trained encoders and untrained encoders to extract deep and shallow features separately, based on the following considerations:
> > > >
> > > > The studies by Zeiler and Fergus (2014) [2] and Yosinski et al. (2014) [3] have demonstrated that shallow and deep features are complementary. Pre-trained encoders (such as CLIP and SimCLR) can extract deep semantic features, which contain strong semantic information and high abstraction, making them well-suited for capturing high-level concepts in images or data. On the other hand, untrained encoders extract shallow visual features, which reflect the local details and structural information of the image. Deep features and shallow features each have their own advantages, and they can complement each other during the label disambiguation process.
> > > >
> > > > In partial label learning (PLL), samples often have multiple candidate labels. To effectively eliminate label ambiguity, the model needs to leverage feature information at different levels. The DAFF module we designed fuses these two types of features using an attention mechanism, allowing the model to fully utilize their complementary information—maintaining local details while enhancing semantic understanding. This fusion not only strengthens the model's stability in label disambiguation but also improves its ability to represent complex data. Additionally, we further strengthen the relationship between labels and features through FLMI. With this mechanism, the model can not only learn the basic association between labels and features but also optimize the matching degree between them, further improving the accuracy of label predictions.
> > > >
> > > > By combining the feature processing strategies of pre-trained and untrained encoders, we provide multi-level feature representations in the label disambiguation process, ensuring that the model has stronger robustness and adaptability in complex PLL tasks. The content has been added to the Introduction.
> > > >
> > > > **W4**
> > > >
> > > > Thank you for the reviewer’s question. We understand that there is a distinction between labels and features: labels are typically discrete and lie within the category simplex, while features are continuous and embedded. Nevertheless, our approach successfully aligns the similarity between labels and features through the FLMI mechanism, ensuring the theoretical and practical justification for this choice.
> > > >
> > > > Firstly, from a theoretical perspective, mutual information quantifies the dependency between two random variables. Although labels are discrete and features are continuous, by using an appropriate distance metric (such as InfoNCE loss), we can map both into a common representation space. Mutual information measures the shared information between features and labels, encouraging their alignment and ensuring that features effectively express the semantic content of the labels. Despite the spatial difference between labels and features, we can learn a mapping between them to maintain consistency. By introducing FLMI, we explicitly maximize the information between labels and corresponding features during training, ensuring that the correlation between label and feature embeddings is preserved. This mechanism helps the labels acquire suitable representations in the feature space, thus improving the label disambiguation process. Additionally, some past research has also explored the similarity between feature embeddings and discrete labels. For example, Suh and Seo (2023) [4] addressed the long-tail classification problem by maximizing mutual information between latent features and true labels (the long-tail classification problem refers to severe class imbalance in real-world datasets, where data for minority classes is scarce, and data for majority classes is abundant). Lee and Kim (2015) [5] associated one feature with two or more labels simultaneously to solve the multi-label classification problem.
> > > >
> > > > Furthermore, in PLL, each sample has multiple candidate labels, with only one being correct. Since we cannot directly access the correct label, the similarity between features and labels becomes one of the solutions for label disambiguation. Although labels are theoretically discrete, by maximizing mutual information between features and labels, we enable the network to learn the “probabilistic” structure of the label space. Thus, even though labels are discrete, features can still effectively express the relative relationships and similarities between labels, helping the model perform correct label classification. Therefore, the mutual information computation between labels and features is grounded in prior theory, and ablation experiments further demonstrate that our FLMI mechanism is effective.

---

> > > > > ### Author Response · Authors · 2025-11-20
> > > > >
> > > > > **W5: Training and inference runtime overheads for the methods aren't reported.**
> > > > >
> > > > > Thank you for the reviewer’s question. We evaluated MDMPLL as well as several comparison methods on an NVIDIA RTX 4090 GPU. On the CIFAR-10 dataset (with $q = 0.3$), the training time for MDMPLL is approximately 4.0 GPU hours, while PICO takes about 3.2 GPU hours, CAVL requires about 2.6 GPU hours, and LS-PLL takes about 2.8 GPU hours. Despite the introduction of iterative diffusion steps, MDMPLL achieves stable convergence within 300 epochs. Moreover, unlike traditional image diffusion models, MDMPLL adds and denoises noise on the labels rather than on the pixels. This design allows our noise prediction network to remain lightweight and efficient, thereby accelerating the training speed for each epoch. Therefore, although MDMPLL incurs higher computational costs, it achieves significantly higher accuracy, making the additional complexity both reasonable and worthwhile.
> > > > >
> > > > > **Q1: Is there a study that demonstrates the effects of the label filtering stage at the beginning?**
> > > > >
> > > > > Thank you for the reviewer’s question. We believe that there is currently no research specifically isolating and theoretically analyzing the initial label filtering phase in PLL under diffusion models. However, our initial label filtering method shares similarities with existing PLL approaches, which smooth candidate labels based on the similarity between features and even labels, thereby improving the robustness and stability of the model. This label filtering is a one-time process and is not a new learning mechanism; rather, it provides relatively cleaner pseudo-labels for subsequent diffusion and label updating. Specifically, combining feature similarity and label similarity can be viewed as performing label propagation on a feature-label graph. This approach helps reduce the variance of initial pseudo-labels, making the subsequent denoising and label updating process less sensitive to noisy labels.
> > > > >
> > > > > The principle behind this is also easy to explain. Features of the same class always tend to cluster together, and although a sample may have multiple candidate labels, only one of them is correct. When samples of the same class cluster together, the number of correct labels increases compared to the other labels in the candidate label set. We can use this to update the candidate label set and obtain the relatively correct label. The smaller the noise rate, the greater the contribution of the label filtering phase. Furthermore, our experiment with the $k$ hyperparameter also serves as a validation for the label filtering phase. Medium-sized values of $k$ typically yield the best performance, while smaller or larger values of $k$ tend to lower the accuracy.
> > > > >
> > > > > [1] Chen, Jian, et al. "Label-retrieval-augmented diffusion models for learning from noisy labels." Advances in Neural Information Processing Systems 36 (2023): 66499-66517.
> > > > >
> > > > > [2] Zeiler, Matthew D., and Rob Fergus. "Visualizing and understanding convolutional networks." European Conference on Computer Vision, 2014, 818-833.
> > > > >
> > > > > [3] Yosinski, Jason, et al. "How transferable are features in deep neural networks?" Advances in Neural Information Processing Systems, vol. 27, 2014.
> > > > >
> > > > > [4] Suh, Min-Kook, and Seung-Woo Seo. "Long-tailed recognition by mutual information maximization between latent features and ground-truth labels." International Conference on Machine Learning, 2023, 32770-32782.
> > > > >
> > > > > [5] Lee, J., and D. W. Kim. “Mutual Information‑based multi‑label feature selection using interaction information.” Expert Systems with Applications, vol. 42, 2015, pp. 2013‑2025.

---

### Author Response · Authors · 2025-11-27

Dear Reviewers,
I hope this message finds you well.As the discussion period is nearing its end ,I wanted to ensure we have addressed all your concernssatisfactorily.If there are any additional points or feedback you'd like us to consider,please let us know.Your insights are invaluable to us,and we're eager to address any remainingissues to improve our work.
Thank you for your time and effort in reviewing our paper.

---

### Meta-Review · Area_Chair_1wG4 · 2026-01-07

**Summary:**

This paper proposes MDMPLL, which adapts diffusion models to partial label learning. The method shows empirical gains over baselines.
Nevertheless, the reviewers raised several significant concerns.

First, the theoretical proofs are largely informal and restate known results, failing to establish method-specific novelty.

Second, the proposed diffusion approach closely follows standard DDPM procedures, with only superficial adaptations for label information.

Third, the suitability of diffusion models for label disambiguation in PLL is questionable and insufficiently justified.

Although the authors attempted to address these concerns in the rebuttal, the responses remain unconvincing. In addition, reviewers raised further issues regarding computational cost and the coverage of related methods, and questioned the claims concerning the joint mechanisms of disambiguation and representation learning, all of which require further attention in future revisions.

Given the consistent reviewer ratings, the current paper remains below the acceptance threshold.

**Reviewer Concerns:**

Some concerns regarding comparisons with LRA-Diffusion, motivation behind label and feature vector similarity, mutual information, ablation results and visualization results have been addressed by the rebuttal. However, a few major concerns  listed above in the summary are not convincingly solved yet.

**Reviewer Scores:**

I do not anticipate major changes over the review scores.

---

### Decision · Program_Chairs · 2026-01-26

Reject